# Beyond the main function: An experimental study of the use of hardwood boomerangs in retouching activities

Eva Francesca Martellotta [1,2]*, Yinika L. Perston [1,2,3], Paul Craft [4,5], Jayne Wilkins [1,2,6], Michelle C. Langley [1,2]

**1** Australian Research Centre for Human Evolution, Griffith University, Brisbane, Queensland, Australia, **2** School of Environment and Science, Griffith University, Brisbane, Queensland, Australia, **3** Griffith Centre for Social Cultural Research, Griffith University, Gold Coast, Queensland, Australia, **4** Birrunburra/Bundjalung/Yugambeh/Yuggera & Turrbal Aboriginal Nations, **5** Burragun Aboriginal Cultural Services, Burleigh Heads, Queensland, Australia, **6** Human Evolution Research Institute, University of Cape Town, Cape Town, South Africa

\* eva.martellotta@griffithuni.edu.au

**Data Availability Statement:** All relevant data are within the manuscript and its Supporting Information files.

## Abstract

Retouched lithic tools result from the functional modification of their edges following knapping operations. The study of the later stages of the reduction sequence is fundamental to understanding the techno-functional features of any toolkit. In Australia, a gap exists in the study of the *chaîne opératoire* of lithic tools shaped or re-shaped through percussion retouching. In our previous works (Martellotta EF., 2021, Martellotta EF., 2022), we have presented evidence for the use of hardwood boomerangs for retouching purposes in Australian Aboriginal communities. Through a detailed experimental protocol, the present study demonstrates how boomerangs can function as retouchers. We found that the use-wear generated on the boomerang's surface during retouch activity is comparable to retouch-induced impact traces observed on Palaeolithic bone retouchers, as well as to experimental bone retouchers generated in our replication experiments. Finally, we explore the role that microscopic lithic chips embedded in the retouchers' surface play in the formation process of retouching marks. Our results address the need for a deeper investigation of percussion retouching techniques in Australian contexts, opening the possibility that uncommon objects —such as boomerangs—could be used for this task. This concept also highlights the broader topic of the highly diverse multipurpose application of many Indigenous tools throughout Australia. At the same time, the study reveals a deep functional connection between osseous and wooden objects—a topic rarely investigated in archaeological contexts.

## Introduction

Retouched lithic tools, and their relative technology, play a crucial role in interpreting the archaeological record of Aboriginal Australia. Studying the final stages of the reduction

**Funding:** EFM was granted financial support from the Griffith University Doctor of Philosophy Program (https://www.griffith.edu.au/study/degrees/doctor-of-philosophy-6001). Funding was also provided by The EXARC Experimental Archaeology Award 2021 to EFM (https://exarc.net/cooperation/ea-award#:~:text=EXARC%20offers%20two%20awards%20of,member%20John%20Kiernan%20(US).). The funders had no role in study design, data collection and analysis, decision to publish, or preparation of the manuscript.

**Competing interests:** The authors have declared that no competing interests exist.

sequence—i.e., retouching and resharpening—is vital to fully understand the functional concepts behind subsistence strategies in Australia's deep and recent past. In previous studies, some retouched lithic industries have received more attention than others [e.g., 1–3]. It resulted in a gap in the current research interest: the absence of a comprehensive investigation of tools shaped by employing percussion retouching techniques. Such investigation should not be limited to the analysis of the *retouched* lithic flakes (i.e., the objective of the retouch activity), but it should also include technological and traceological studies of the *retouching* tools (i.e., the means used to carry out the retouch activity).

In our previous works, preliminary investigating this issue [4, 5], we proposed the use of hardwood boomerangs for retouching purposes among Indigenous Australian communities until at least European incursions. Our hypothesis—and the foundation of the present experiment—was based on four central notions: (1) boomerangs hold a deep multipurpose value in Australian Aboriginal societies [e.g., 5–9]; (2) literature evidence proved that the use of boomerangs (and other wooden tools) in retouching and resharpening activities could be technologically compared to the use of Palaeolithic bone retouchers [5]; (3) a traceological analysis of museum-curated boomerangs identified the presence of retouch-induced impact traces comparable with the ones identified on bone retouchers [4]; (4) bone and wood, as materials for tool-making, share some mechanical and physical properties, and it is possible to hypothesise they would have similar reactions to the percussion movement in retouching activities.

The latter of these points is, at the moment, only a fair assumption based on the similar cellular structure of osseous and wooden materials, which gives them a certain degree of density, hardness, and elasticity [10–14]. In Australian archaeology, few studies have investigated potential similarities, technologically speaking, between osseous and wooden objects [12, and references therein]. This lack of studies is mainly owing to the rarity of wooden items recovered in archaeological contexts because of harsh environmental conditions [14, 15, and references therein]. On the other hand, most of the knowledge of Australian wooden tools comes from ethnographic sources. Among these, evidence of wooden implements—especially boomerangs—employed in percussion retouching of lithic tools is rare but present [5, 6, 8, 9, 16–23]. However, none of those contributions engages in an experimental protocol aimed at investigating the technology behind the use of boomerangs in retouching activities. An exception is [20]: one of the few recent contributions addressing, experimentally, the topic of boomerangs used to shape stone tools; this work, however, mainly focused on the retouched stone tools (Tula adze) rather than on the boomerangs *per se*. As a result, the technological implications of the use of wooden boomerangs in retouching activities constitute a gap in the current research. If and how those implications could be compared with osseous tools used as retouchers is an even less investigated topic.

The present work aims to fill this gap. We present a structured experimental program to understand the use of hardwood boomerangs as retouching tools. The questions addressed with this experiment are (1) which variables are involved in the use of boomerangs to retouch lithic tools? (2) how does the resulting use-wear compare to that observed on bone retouchers? We expect to easily reproduce, using boomerangs, intensively retouched lithic flakes reflecting morphologies found in the archaeological records. We aim to formulate a detailed description of retouch-induced use-wear on boomerangs. Based on our traceological and literature studies [4, 5], we also expect similarities between the impact traces of bone and boomerang retouchers in terms of morphology, distribution and internal cross-section. This work enriches methodologies applied to the traceological study of boomerangs, which have so far only been marginally investigated [15, 24, 25]. Furthermore, our results could be used as a reference in the analysis of technological similarities of osseous and wooden implements in archaeological investigations of both Aboriginal Australia and Palaeolithic Europe.

## Materials and methods

The experimental program involved four stages: manufacturing lithic flakes, manufacturing bone retouchers and hardwood boomerangs, retouching lithic flakes with bone retouchers, and retouching lithic flakes with boomerangs. Subsequent to the retouching sessions, the resulting use-wear was analysed and compared by creating a database and applying basic statistic tests using the software R [26]. In the measurements of some specific use-wear, the 'sample' function in R software was used to generate a random subset of data, more suitable for statistic correlations. Pictures and videos of the experimental materials and sessions were recorded using the following equipment: a Canon PowerShot SX400 IS digital camera, a GoPro HERO7 White camera (v. 02.10) and a Canon EDS 800D digital SLR camera.

### Manufacturing lithic flakes

The experimental lithic flakes were made from two cobbles of Texas chert shipped from Meadville (Pennsylvania, USA) by the service 'Flintknapping Supplies' (https://flintknappingsupplies.com/). The lithic raw material was chosen based on its petrographic and lithological characteristics, comparable with flints recovered in Palaeolithic European sites and some of the finer cherts identified in Australian contexts. We are aware of the greater lithological variety present in the Australian continent; however, we decided to limit this experiment to only one type of lithic raw material to avoid unnecessary variables interfering with the use-wear analysis on the boomerangs. Although previous experimental studies of bone retouchers revealed some differences in the features of the use-wear when retouching either coarse or fine lithic materials [27], we believe it is a statistically and qualitatively not a relevant issue in the context of our study.

The flaking of the two cobbles ("cobble_1", S1AA Fig and "cobble_2", S1AB Fig) was carried out by an expert knapper (Y.L.P.) using two large, fine-grained basalt hammerstones, which weigh 570 g and 700 g, respectively. The flakes produced were identified with the letter F followed by sequential numbers. These flakes were measured according to their debitage axis (length, width, maximum thickness, all expressed in millimetres; S2A Fig) and weighed, before being retouched. The cross-section morphology of each side of the flake (proximal, distal, right, left) was recorded, as well as the measures of the edges' angles, following [28] (S1B Fig). Finally, a rough percentage of cortex covering the flake surface was recorded (S1 Table).

### Manufacturing bone retouchers

The Palaeolithic bone retouchers were produced mainly by breaking ungulate long bones. Depending on geographical and chronological factors, bone retouchers are obtained from different animal species, most commonly Cervidae and Bovidae, but also Carnivora and, in a few instances, humans [29–34, among others; for an overview, see contributions in 35]. In this experiment, we used forelimbs and hindlimbs (N = 8) of two sub-adult female individuals of *Bos taurus*, purchased from a local butcher in Brisbane (Australia). Each bone was identified with the letter B followed by a progressive number and then cleaned (i.e., tendon removal) using a metal scalpel. Only one tibia (B03) was subject to disarticulation operations finalised at the removal of the epiphysis. The periosteum was not removed during the cleaning process but only when it constituted an obstacle to breaking activities. After the cleaning, maximum length, width, thickness, circumference, and weight were recorded in millimetres and grams, respectively (S3A Fig).

The breaking activity (S3B and S3C Fig) took place in the ARCHE Archaeology Lab at Griffith University. It involved two operators—one experienced experimental zooarchaeologist (E. F.M.) and one unexpert zooarchaeologist. The operators applied a direct percussion technique

**Table 1. Bone retouchers used during retouching session 1 (N = 14).**

| Retoucher ID | Animal species | Skeletal element | Bone laterality | Bone ID | Periosteum removal | Max. length (mm) | Max. width (mm) | External thickness (mm) | Weight (g) | N. of use areas |
|---|---|---|---|---|---|---|---|---|---|---|
| R1 | *Bos taurus* | tibia | L | B03 | yes | 98.2 | 28.8 | 22.1 | 42.5 | 1 |
| R6 | *Bos taurus* | tibia | L | B03 | no | 152.5 | 34.8 | 19.6 | 66.1 | 2 |
| R7 | *Bos taurus* | tibia | R | B05 | yes | 103.3 | 48.9 | 28.9 | 60.3 | 2 |
| R8 | *Bos taurus* | tibia | R | B05 | yes | 110.8 | 28.2 | 20.7 | 43.4 | 2 |
| R9 | *Bos taurus* | tibia | R | B05 | yes | 119.5 | 29.4 | 29.9 | 89.8 | 1 |
| R10 | *Bos taurus* | tibia | L | B03 | no | 107.6 | 37 | 26.9 | 52.8 | 2 |
| R11 | *Bos taurus* | femur | R | B07 | yes | 155 | 46.6 | 25.4 | 146.1 | 3 |
| R12 | *Bos taurus* | femur | R | B07 | yes | 108 | 42.4 | 20.1 | 55 | 1 |
| R15 | *Bos taurus* | radius/ulna | R | B02 | no | 85.4 | 31.8 | 7.8 | 17.7 | 2 |
| R17 | *Bos taurus* | femur | L | B04 | yes | 195 | 48.6 | 23 | 153 | 1 |
| R18 | *Bos taurus* | femur | L | B04 | yes | 150 | 48.3 | 21 | 78.2 | 1 |
| R19 | *Bos taurus* | humerus | R | B06 | yes | 89 | 46.7 | 25 | 64.4 | 2 |
| R20 | *Bos taurus* | humerus | R | B06 | yes | 96.4 | 44.6 | 27.6 | 51.2 | 1 |
| R21 | *Bos taurus* | humerus | R | B01 | yes | 89.5 | 48.6 | 23.4 | 61.8 | 1 |

using two large, fine-grained hammerstones; in three instances, an anvil was used to facilitate the breaking; finally, two flakes were occasionally used to remove the periosteum if it was impeding the breaking activity. A detailed description of the breaking session is present in S2 Table and S1 File. In total, 74 bone blanks resulted from the breaking: 23 of them were selected by E.F.M. to be used as retouchers based on morphometric criteria and similarities with archaeological examples; the rest (N = 51; weight = 238.1g) were discarded because they were considered not suitable for retouching activities.

Of the 23 obtained retouchers, identified with the letter R and a sequential number, 14 were used in the retouching sessions (Table 1). Of these, the majority were of tibia (N = 6), whereas a smaller percentage were femurs (N = 4), humeri (N = 3) and radii/ulnae (N = 1). These data find good consistency with previous zooarchaeological studies on Palaeolithic bone retouchers assemblages [35]. General measurements of the retouchers include: maximum length and width recorded according to the major axis of the bone blank; weight; external thickness; cortical bone thickness (S2B Fig). The width of the arc (C) and the height of the midpoint of the arc (H) were measured in correspondence to the use area. Those measurements are necessary to calculate the radius of the arch to assess the role that the convexity of the surface plays in the retouch activity. The radius of the arch was calculated as follows: $H/2+C^*2/8^*H$ [36]. More detailed information on the retouchers can be found in Table 1 and S1 File.

## Hardwood boomerangs

Before the first European incursions, Aboriginal peoples across Australia applied various methods and techniques to manufacture boomerangs, most commonly using stone tools. Boomerangs were made from different parts of the tree, including tree trunks, elbow bend branches and tree roots. At the first stage of manufacturing, a hafted stone axe was used to retrieve the desired section of the tree. During the following stages, stone or shell scraping tools such as adzes were used to create a pre-form. The next step could either involve the hardening of the boomerang over hot coals or its soaking and twisting in water—depending on the sought-after aerodynamic features. Finally, the surface was sanded smooth, often with sandpaper fig leaves. After the arrival of Europeans, Australian Aboriginal peoples had access to new materials (e.g., steel tools) which proved to be popular as they resulted in a faster, easier and

more efficient way to shape wooden tools [some recent contributions to the manufacturing process could be found in 5, 7, 15, 24, 37, and references therein].

Four hardwood boomerangs were used in the present experiment, identified with the letter B and a sequential number, and manufactured by two expert Indigenous Australian artisans. These boomerangs were made as usable weapons using modern steel tools, including tomahawk, rasp files, sandpaper and electric 4-inch grinders. These modern techniques mimic, but greatly enhance, the speed and efficiency of the traditional methods used by the two artisans' ancestors. As per experimental protocol, a screening of the manufacturing marks on the boomerangs' surface has been carried out before the retouching sessions to ensure a proper distinction between manufacturing marks and use-wear.

B1 (S4A Fig) and B2 (S4B Fig) are handcrafted by P.C., a Birrunburra Bundjalung Yugambeh Yuggera Turrbal man of Southeast Queensland and Northeast New South Wales. In the traditional language of this area, *bargan* or *burragunn* are the words used to indicate a returning (symmetrical) boomerang, whereas the asymmetrical type is called *baring*. Millmullian– Laurence Magick Dennis, a Wailwaan and Yuin man from the Southeast of New South Wales, handcrafted B3 (S4C Fig) and B4 (S4D Fig). The traditional languages of this area have several words for "hunting boomerang": *bubarra* (Gamilaraay/Yuwaalaraay language), *garrbaa* (Wiradjuri language) or *biyarr* (Wailwaan language). For the sake of clarity, the English term "boomerang" will be used henceforth to refer to those tools.

B1 is a symmetrical boomerang made from black wattle wood (*Acacia mearnsii*) and painted with acrylic colours. The paint was scraped from one part of the boomerang's surface to test its potential interference with the observation of the retouch-induced use-wear. B2 is an asymmetrical boomerang (or hunting boomerang) made from ironbark wood (*Eucalyptus* sp.); its surface has been varnished. They both have a plano-convex cross-section. B3 and B4 are asymmetrical boomerangs handcrafted from mulga wood (*Acacia aneura*); their surface was oiled with vegetable-based oil, and they both have a bi-convex cross-section.

The general measurements of the boomerangs were recorded as shown in S2D Fig. The maximum length is measured from tip to tip, whereas the maximum width is measured in correspondence of the elbow. Among our materials, lengths vary between 560 and 650 mm, and widths between 59 and 67 mm; the thickness is between 14 and 16 mm, whereas the weight varies between 335 and 405 g (Table 2). Finally, the radius of the arch was calculated as showed above, in order to assess the convexity of the boomerang's surface in correspondence of the retouch-induced use-wear.

### Retouching sessions

Using bone retouchers—The first retouching session (S2 and S3 Files), involving bone retouchers, was carried out in the ARCHE Archaeology lab by two experienced knappers (Y.L. P. and Tim R. Maloney). The operators freely choose which retouchers and lithic flakes to use, resulting in 14 bone retouchers used to shape the same number of flakes (Table 3). The operators aimed to produce flakes that they felt might be functional for cutting or scraping activities,

**Table 2. Details of boomerangs used during retouching session 2.**

| Tool ID | Surface modification | Boomerang type | Max. length (mm) | Max. width (mm) | Max. thickness (mm) | Weight (g) | Elbow internal angle (°) | N. of use areas |
|---|---|---|---|---|---|---|---|---|
| **B1** | paint | symmetrical | 560 | 59.2 | 14.3 | 405 | 174 | 4 |
| **B2** | varnish | asymmetrical | 655 | 63.3 | 14.4 | 335 | 170 | 4 |
| **B3** | vegetal oil | asymmetrical | 620 | 65.8 | 15.3 | 343 | 169 | 5 |
| **B4** | vegetal oil | asymmetrical | 630 | 67.2 | 16 | 385 | 169 | 4 |

**Table 3. Analysis of the use areas on bone retouchers and boomerangs.**

| Tool ID | Metric data of use areas | | | | | | | | | | | | | | impact traces analysis | | | | | |
|---|---|---|---|---|---|---|---|---|---|---|---|---|---|---|---|---|---|---|---|---|
| | Knapper | N. of blows | Length (mm) | Width (mm) | Surface (mm²) | Circumference (mm) | Cortical bone thickness (mm) | H value / Thickness (mm) | C value (mm) | Radius of the Arch (mm) | Linear impressions | Punctiform impressions | Striations | Notches | Scraping marks | Intensity | Average Length Linear I. (mm) | Average Surface Punctiform I. (mm²) | Average Circumference Punctiform I. (mm) | Retouched flake ID |
| R1 | Y.L.P. | 136 | 25.9 | 21.0 | 366.4 | 92.1 | 8 | 22.5 | 31.2 | 11.6 | 114 | 18 | 0 | 1 | yes | S | 1.0 | 0.07 | 1.04 | F07 |
| R6_1 | Y.L.P. | 28 | 25.9 | 10.0 | 165.3 | 80.7 | 8 | 12.3 | 28.5 | 6.7 | 249 | 38 | 0 | 1 | yes | S | 1.0 | 0.04 | 0.79 | F74 |
| R6_2 | Y.L.P. | 61 | 52.9 | 12.9 | 276.1 | 125.9 | 11.4 | 14.6 | 26.2 | 7.7 | 106 | 7 | 0 | 2 | yes | S | 1.3 | 0.08 | 1.06 | F74 |
| R7_1 | T.R.M. | 7 | 12.3 | 30.0 | 157.2 | 88.2 | 10 | 11.2 | 26.4 | 6.2 | 14 | 0 | 0 | 0 | no | D | 1.5 | 0.00 | 0.00 | F37 |
| R7_2 | T.R.M. | 53 | 39.6 | 20.7 | 397.6 | 133.8 | 11 | 25.7 | 29 | 13.1 | 118 | 8 | 0 | 0 | no | S | 1.3 | 0.11 | 1.25 | F37 |
| R8_1 | Y.L.P. | 189 | 7.9 | 6.1 | 237.4 | 22.9 | 7.6 | 15.4 | 27.4 | 8.1 | 165 | 13 | 2 | 3 | yes | C | 1.0 | 0.06 | 0.90 | F50 |
| R8_2 | Y.L.P | 9 | 29.3 | 22.0 | 362.0 | 103.7 | 11 | 11 | 27 | 6.1 | 31 | 0 | 0 | 0 | no | C | 0.9 | 0.00 | 0.00 | F50 |
| R9 | T.R.M. | 59 | 43.4 | 20.6 | 529.1 | 201.9 | 11 | 29.4 | 30.5 | 15.0 | 432 | 69 | 4 | 4 | yes | S | 0.9 | 0.05 | 0.83 | F55 |
| R10_1 | T.R.M. | 4 | 25.2 | 26.2 | 249.7 | 93.6 | 20.8 | 23.6 | 34 | 12.2 | 31 | 27 | 0 | 0 | no | D | 1.0 | 0.06 | 0.90 | F78 |
| R10_2 | T.R.M. | 46 | 16.4 | 16.0 | 131.9 | 57.8 | 12 | 35.6 | 34.8 | 18.0 | 9 | 0 | 0 | 1 | no | C | 1.0 | 0.00 | 0.00 | F78 |
| R11_1 | T.R.M. | 12 | 17.6 | 29.6 | 285.9 | 85.7 | 7 | 23 | 43.4 | 12.0 | 4 | 0 | 0 | 1 | yes | I | 1.6 | 0.00 | 0.00 | F54 |
| R11_2 | T.R.M. | 25 | 21.0 | 28.0 | 245.1 | 71.1 | 15 | 15 | 44 | 8.2 | 21 | 0 | 1 | 0 | no | D | 1.3 | 0.00 | 0.00 | F54 |
| R11_3 | T.R.M. | 31 | 11.7 | 13.4 | 771.4 | 43.4 | 8 | 8 | 18 | 4.6 | 28 | 0 | 1 | 1 | no | C | 0.8 | 0.00 | 0.00 | F54 |
| R12 | Y.L.P. | 50 | 38.2 | 29.0 | 650.6 | 116.9 | 10 | 21.1 | 40.2 | 11.0 | 154 | 7 | 2 | 1 | yes | S | 1.5 | 0.04 | 0.77 | F47a |
| R15_1 | Y.L.P. | 27 | 32.9 | 25.9 | 426.3 | 102.3 | 5 | 4.6 | 29.6 | 3.9 | 43 | 31 | 1 | 0 | yes | C | 1.0 | 0.03 | 0.68 | F06 |
| R15_2 | Y.L.P. | 76 | 12.8 | 24.3 | 114.2 | 74.8 | 4 | 5.6 | 28.6 | 4.1 | 23 | 11 | 0 | 0 | no | D | 0.9 | 0.04 | 0.82 | F06 |
| R17 | T.R.M. | 63 | 39.2 | 23.0 | 274.1 | 108.4 | 8 | 13 | 39 | 7.3 | 42 | 12 | 0 | 0 | no | C | 1.0 | 0.04 | 0.74 | F05 |
| R18 | T.R.M. | 9 | 21.2 | 16.3 | 176.7 | 103.8 | 11 | 11.3 | 39 | 6.5 | 79 | 6 | 1 | 0 | yes | D | 1.1 | 0.03 | 0.64 | F25 |
| R19_1 | T.R.M. | 28 | 25.6 | 22.2 | 254.0 | 95.1 | 16 | 24.4 | 43 | 12.6 | 45 | 0 | 0 | 0 | yes | D | 0.8 | 0.00 | 0.00 | F58 |
| R19_2 | T.R.M. | 25 | 21.5 | 15.9 | 151.8 | 67.1 | 8 | 8.5 | 42 | 5.5 | 51 | 0 | 1 | 1 | yes | D | 0.8 | 0.00 | 0.00 | F58 |
| R20 | Y.L.P. | 100 | 36.6 | 22.6 | 375.4 | 154.4 | 20 | 25.3 | 43.6 | 13.1 | 148 | 5 | 0 | 1 | no | C | 1.0 | 0.05 | 0.86 | F30 |
| R21 | Y.L.P. | 118 | 36.4 | 25.1 | 448.6 | 128.3 | 13.4 | 19.3 | 43.8 | 10.2 | 208 | 4 | 2 | 2 | yes | C | 0.8 | 0.04 | 0.76 | F39 |
| B1_1 | Y.L.P. | 109 | 34.13 | 33.74 | 824.26 | 211.61 | NA | 13.2 | 49.2 | 7.53 | 265 | 13 | 0 | 2 | yes | S | 1.53 | 0.13 | 1.42 | F56 |
| B1_2 | Y.L.P. | 98 | 27.61 | 23.38 | 508.28 | 114.88 | NA | 13.2 | 48.5 | 7.52 | 219 | 67 | 0 | 0 | no | C | 1.09 | 0.04 | 0.67 | F59 |
| B1_3 | Y.L.P. | 105 | 44.5 | 35.4 | 1047.0 | 224.0 | NA | 11.7 | 57.7 | 7.08 | 211 | 43 | 0 | 3 | no | C | 2.18 | 1.21 | 1.42 | F51 |
| B1_4 | Y.L.P. | 103 | 97.2 | 38.0 | 2193.0 | 400.7 | NA | 12.8 | 58.6 | 7.54 | 264 | 46 | 5 | 2 | yes | D | 1.64 | 0.17 | 1.48 | F90 |
| B2_1 | Y.L.P. | 104 | 58.5 | 34.0 | 1342.8 | 334.5 | NA | 12.4 | 63.4 | 7.48 | 347 | 109 | 1 | 2 | yes | C | 1.00 | 0.03 | 0.66 | F100 |
| B2_2 | Y.L.P. | 129 | 41.6 | 41.2 | 952.5 | 257.3 | NA | 13.6 | 62.2 | 7.94 | 353 | 112 | 1 | 2 | yes | C | 1.38 | 0.04 | 0.76 | F71 |
| B2_3 | Y.L.P. | 69 | 46.5 | 27.9 | 974.4 | 268.0 | NA | 10.3 | 35.2 | 6.00 | 427 | 97 | 1 | 2 | yes | S | 1.16 | 0.03 | 0.67 | F17 |
| B2_4 | Y.L.P. | 26 | 32.3 | 30.8 | 500.2 | 230.0 | NA | 10.8 | 35.8 | 6.23 | 277 | 76 | 1 | 1 | yes | C | 1.10 | 0.04 | 0.81 | F83 |
| B3_1 | Y.L.P. | 63 | 29.1 | 19.3 | 319.2 | 149.3 | NA | 9.9 | 30.2 | 5.71 | 257 | 67 | 2 | 2 | yes | C | 1.34 | 0.09 | 1.09 | F36 |
| B3_2 | Y.L.P. | 81 | 35.9 | 21.8 | 457.7 | 234.0 | NA | 10.5 | 32.6 | 6.03 | 559 | 70 | 2 | 0 | yes | C | 0.83 | 0.05 | 0.88 | F73 |
| B3_3 | Y.L.P. | 50 | 96.9 | 19.7 | 1121.0 | 435.4 | NA | 12.0 | 35.3 | 6.74 | 309 | 112 | 1 | 0 | yes | D | 0.98 | 0.04 | 0.79 | F03 |
| B3_4 | Y.L.P. | 145 | 50.9 | 29.3 | 811.7 | 271.7 | NA | 14.1 | 57.6 | 8.07 | 452 | 209 | 6 | 1 | yes | C | 1.19 | 0.06 | 0.91 | F76 |
| B3_5 | Y.L.P. | 61 | 29.4 | 43.7 | 721.0 | 250.5 | NA | 14.0 | 61.3 | 8.09 | 319 | 111 | 0 | 2 | yes | S | 1.20 | 0.06 | 0.91 | F40 |
| B4_1 | Y.L.P | 29 | 36.4 | 23.4 | 487.5 | 217.4 | NA | 16.1 | 35.8 | 8.61 | 450 | 89 | 2 | 2 | yes | C | 1.20 | 0.06 | 0.91 | F60 |
| B4_2 | Y.L.P | 45 | 41.1 | 21.1 | 464.7 | 221.9 | NA | 15 | 34.7 | 8.08 | 491 | 171 | 3 | 1 | yes | C | 0.96 | 0.04 | 0.79 | F35 |
| B4_3 | Y.L.P. | 6 | 39.3 | 36.7 | 606.7 | 311.7 | NA | 15 | 60.4 | 8.51 | 694 | 173 | 4 | 0 | yes | D | 1 | 0.05 | 0.83 | F81 |
| B4_4 | Y.L.P. | 80 | 32.2 | 31.8 | 575 | 203.7 | NA | 16 | 61.6 | 8.96 | 467 | 86 | 2 | 0 | yes | C | 0.88 | 0.04 | 0.76 | F41 |

Multiple use areas on the same tool are indicated with underscore symbol and ascending number following the Retoucher ID (example: R10_1). 'NA' stands for "Not Applicable'. In the 'Intensity' column, 'C' = 'concentrated', 'S' = 'superposed', 'D' = 'dispersed', 'I' = 'isolated'.

inspired by retouched tools from Palaeolithic contexts [e.g., Discoid and Quina Mousterian: 31, 32, and references therein]. The retouch activity was carried out by applying a percussion movement following an elliptical trajectory with a tangential point of impact. The knappers did not have any limitations in terms of time or number of blows struck with the retoucher; each activity aimed to retouch the margins of the flake to the knapper's arbitrary satisfaction.

Using boomerangs as retouchers—The second retouching session (Fig 1; S4 File) involved the use of hardwood boomerangs and was carried out at the Griffith Experimental Archaeology Research Facility (GEAR) by an experienced knapper (Y.L.P.), who, however, never used a boomerang to retouch lithic tools. The operator aimed to produce flakes potentially functional for cutting or scraping activities, inspired by retouched tools from Australian archaeological contexts [e.g., 38]. Because there are no recent experimental protocols applicable to the use of boomerangs as retouching tools, the applied movement was inspired by the use of bone retouchers—i.e., direct percussion following an elliptical trajectory with a tangential point of impact. This methodological decision was based on several similarities identified in the literature, regarding the retouching movement applied to boomerangs or bone retouchers [5]. Moreover, part of the experimental protocol aimed at verifying a potential connection between

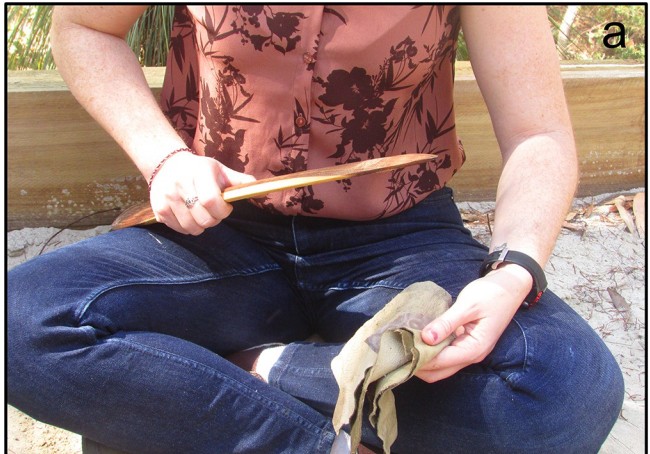
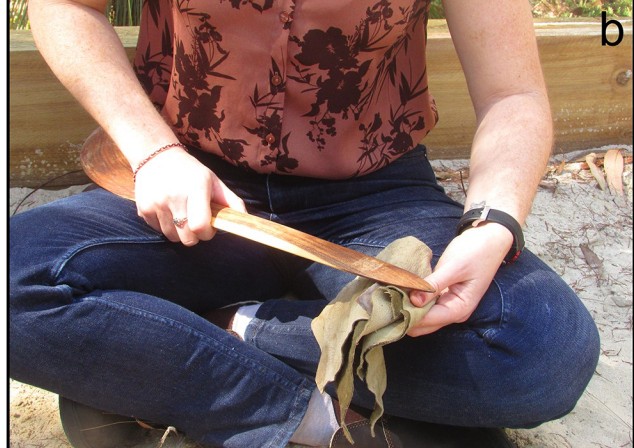
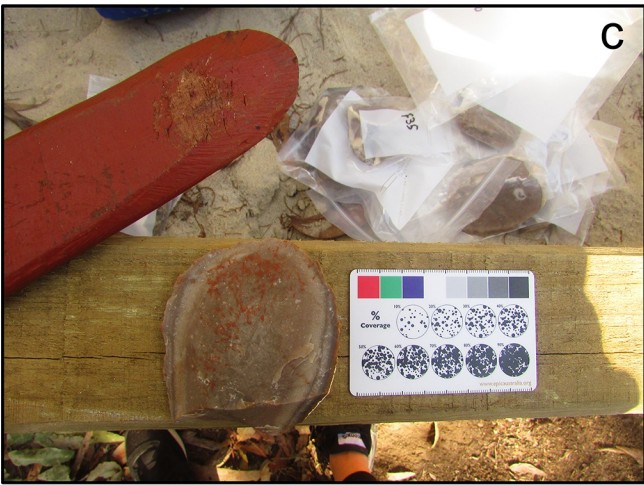
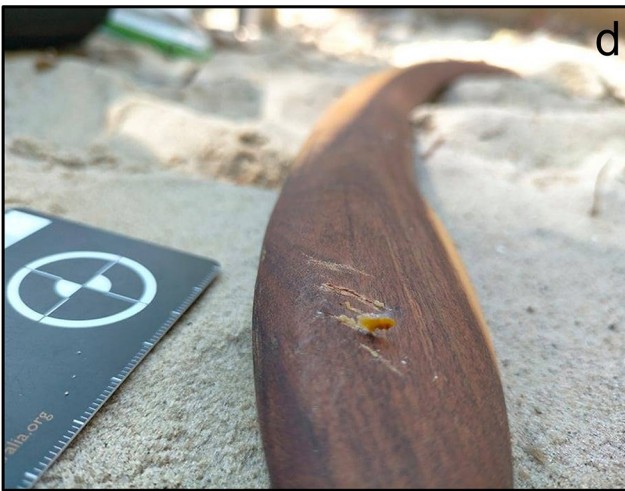

**Fig 1. Retouching session 2 using boomerangs.** (a, b) tangential movement of retouch; (c) use area B1_1 and retouched lithic flake F56; as this boomerang is painted, the ventral face of the flake shows residues of paint oriented according to the percussion movement of retouch; (d) flint chip embedded within retouch impact traces at the initial stage of retouch.

the spatial distribution and the shape of boomerangs. Based on our previous traceological study of museum-curated boomerangs [4], we hypothesised that variations in the boomerangs' morphologies could influence their grasping during the retouch activity (S5 Fig), i.e., variations in the retouch movement applied by the operator.

Finally, we defined the functionality of boomerangs used as retouchers based on (1) the description of the obtained retouched lithic edges, (2) feedback collected from the operator regarding effectiveness and comfortability, and (3) traceological similarities between retouch-induced impact traces observed on boomerangs and bone retouchers.

Both sessions were documented in writing; feedback from the operators was collected, and it is presented where relevant. The feedback included (1) level of comfort in grasping and using the retouching tool, (2) technical objectives, i.e., the aim of retouching, (3) problems encountered during the retouch activity, and (4) solutions offered by the operator to complete the retouch. Each retouched flake was analysed after the retouching sessions. The modified edges were described following [39]: the considered features concern the position of the retouch, the delineation of the lithic edge after the retouch, the morphology of the detachments, the distribution of the retouch on each edge, the extent of the detachment scars on the flake's surface, and the edge-angle of the lithic margin after the retouch (S1C Fig).

## Retouch-induced impact traces analysis

The use-wear resulting from the impact between the lithic edge and the osseous/wooden retouching tool's surface is defined as "retouch-induced impact traces"; these traces group in small portions of the surface of the retouchers, defined as "use-areas" [27, 33]. In our experimental sample, the impact traces were counted and grouped into four morphological categories based on previous works on bone retouchers [27, 33, 40]: (1) linear impressions are elongated, deep marks; (2) punctiform impressions are triangular or ovoidal depressions; (3) striations are short, shallow, and often parallel marks; (4) notches are deep and wide detachments of a small portion of the organic surface during an intense retouch activity. Finally, scraping marks can be present: they appear as long, shallow, linear marks covering a significant portion of the surface. The four morphologies of impact traces usually appear together, often overlapping, in the same use area. Their interaction generates four categories of "intensity of retouch": isolated, dispersed, concentrated, and superposed [27, 33]. In our previous traceological study on boomerangs [4], we proposed that those use-wear, with similar features, also occur on boomerangs used in retouching activities. To verify this traceological evidence, we compared retouch-induced use-wear on boomerangs with impact traces on bone retouchers in our experimental sample.

In order to identify and classify the use-wear, we studied both osseous and wooden tools using low-to-high-powered magnifications of the use areas under an Olympus DSX10-UZH optical microscope, alternating the use of three lenses (DSX10-SXLOB plan 1x/0.03; DSX10-SXLOB plan 3x/0.09; DSX10-XLOB plan FL 10x/0.30) depending on the type of use marks. To reach a higher level of understanding of the impact traces, we performed 3D scans of each morphological category at a micrometric scale. The resulting images have been produced with the Olympus DSX1000 software (v. 1.1.5.13). The analysis of the 3D images was based on the following criteria: cross-section, depth, measurement of the cross-section area, length for linear impressions, surface and perimeter for punctiform impressions. Because of the high number of marks, we selected samples of each category proportional to their presence in the general traceological assemblage. We measured the cross-section area of each stigma; these sections were obtained by slicing the 3D scan of the stigma and moving a plane along its major axis, recording each section at regular intervals.

## Results

### Flake production

The knapping session produced 66 flakes from cobble_1 and 49 flakes from cobble_2; a total of 114 usable flakes (S1AC and S1AD Fig and Table 1). The knapping of cobble_1 produced 91.2 g of knapping débris (i.e., shattered fragments), and the exhausted core bears cortex on 20% of its surface (S1AE Fig). The knapping of cobble_2 produced 61.9 g of débris, and the exhausted core shows no cortex on its surface (S1AF Fig). The resulting flakes varied in size from 22 to 144 mm in length and 17 to 131 mm in width. The majority of the flakes had an abrupt proximal cross-section and a bi-plan distal cross-section, showing angles between 20˚ and 65˚ wide; right and left lateral margins mostly showed a bi-plan cross-section and variable angles (S1 Table).

### Retouching session 1—Bone retouchers

The length of the utilised retouchers ranges between 85.4 mm and 195 mm (average: 118.6 mm), and the width ranges from 28.2 mm to 48.9 mm (average: 40.33 mm). The external thickness ranges from 7.8 and 29.9 mm (average: 27.8). Finally, the weight ranges from 17.7 g to 153 g (average: 70.2 g) (Table 1). The cortical thickness in correspondence to each use area could be grouped in two metric categories: 5–10 mm (41%) and 10–15 mm (36%). The radius of the arch varies from 3.91 to 18.5 mm (average = 9.26 mm; SD = 3.83); most of the use areas (N = 18) locate on a portion of the bone where the radius of the arch measures between 5 and 15 mm (Table 3).

Each retouch activity lasted two minutes on average. The number of blows ranges from a minimum of 4 to a maximum of 189 blows per retoucher (average: 52 blows). In half of the cases (N = 7), only one portion of the bone surface was used; two portions were used on six retouchers, and in only one case, three portions of the surface were exploited (Table 3). The operators chose to switch to a different portion of the surface when the first chosen one was unsuitable for retouching (R7, R10, R11, R15) or because the operator wanted to try more than one surface (R6, R8, R19). Finally, the retouch was interrupted in one case (R18) because the flake (F25) broke.

A complete information set of the entire sample is presented in Table 3 and S3 File; data on the retouched lithic flakes are presented in Table 4.

### Retouching session 2—Hardwood boomerangs

Each retouching activity using boomerangs lasted two minutes on average; a minimum of 6 and a maximum of 145 blows were struck (average: 86 blows). The thickness of the boomerangs, in correspondence with the use areas, measures 13 mm (σ = 0.7) on average, whereas the radius of the arch results in 7.4 mm on average (σ = 0.98).

Based on the technique applied in the use of bone retouchers and on the literature evidence on boomerangs used in retouching activities, the knapper performed the percussion retouch with boomerangs by applying an elliptical trajectory, with a tangential point of contact between the wooden surface and the lithic edge (Fig 1). Although the movement follows the same technique as bone retouchers, the ellipse of the trajectory applied in this session seems to have a smaller diameter, causing the movement to look shorter (S4 File). Data on this retouching session are in Table 3.

B1 was exploited in four portions of its surface, each used to retouch one lithic flake (Fig 2). Three of these use areas are located near the tip, whereas the use area 4 is on the central portion of one of the arms. All the use areas were exploited with approximately 100 blows. This

**Table 4. Details of lithic flakes retouched in session 1 and session 2.**

| Tool ID | Knapper | Max. length (mm) | Max. Width (mm) | Thickness (mm) | Weight (g) | Position | Delineation | Morphology | Localisation | Distribution | Extent | Angle | Proximal Cross-Section | Distal Cross-Section | Right Cross-Section | Left Cross-Section | Proximal Angle | Distal Angle | Right Angle | Left Angle | Cortex |
|---|---|---|---|---|---|---|---|---|---|---|---|---|---|---|---|---|---|---|---|---|---|
| R1 | Y.L.P. | 61 | 50 | 18 | 57.6 | dir (R, L); inv (P) | rec (R); conv (D, L) | ste | D; R; L | cont (R); part (L, D) | sho | ab | ab | p/conv | b/p | p/conv | >90 | 60 | 75 | 60 | <20% |
| R6 | Y.L.P. | 66 | 63 | 24 | 105.5 | dir | conv | ste | D; R; L | cont | sho | ab (D, R); sa (L) | p/conv | p/conc | b/p | p/conc | 60 | 55 | 75 | 60 | 20–50% |
| R7 | T.R.M. | 94 | 49 | 25.8 | 100.8 | dir | rec (L); conv (R) | sca (R); ste (L) | R; L | cont (L); part (R) | sho | sa | ab | ab | b/p | p/conc | 90 | >90 | 60 | 55 | 20% |
| R8 | Y.L.P. | 127.3 | 90.6 | 25 | 374.5 | dir | rec (R); conv (L, D) | par (R); sca (L); sp (D) | R; L; D | cont | lon | ab (L, D); sa (R) | ab | b/p | p/conc | p/conc | 90 | 60 | 55 | 70 | 20–50% |
| R9 | T.R.M. | 97 | 65 | 31.7 | 222.8 | dir | den | ste | D; R; L | cont | inva | ab (D); sa L, R) | ab | p/conc | p/conc | p/conc | 90 | 65 | 75 | 75 | >50% |
| R10 | T.R.M. | 62 | 53 | 11 | 61.9 | dir | den | sca | D; R | cont | lon | low | ab | p/conc | p/conc | p/conc | 90 | 45 | 60 | 80 | >50% |
| R11 | T.R.M. | 64.4 | 92.4 | 30 | 180.8 | bif | irr | ste | R; L; D | cont | sho (dorsal); inva (ventral) | low | ab | b/p | b/conc | b/p | 90 | 85 | 85 | 60 | >50% |
| R12 | Y.L.P. | 76 | 32.4 | 13.8 | 39 | dir | rec (R); conv (D) | sca (R); par (D) | D; R | cont | sho | low | ab | p/conv | b/p | ab | >90 | 55 | 50 | 90 | 20–50% |
| R15 | Y.L.P. | 72 | 44 | 12.8 | 67.4 | dir | rec (R); conv (L); poin (D) | sp | D; R; L | cont | sho | sa | ab | p/conc | p/conc | p/conc | >90 | 80 | 55 | 65 | 20% |
| R17 | T.R.M. | 42.4 | 26.6 | 12.5 | 13.3 | dir | den | sca | D; L | cont | sho | sa | ab | p/conc | p/conc | p/conc | 90 | 65 | 55 | 45 | <20% |
| R18 | T.R.M. | NA | NA | NA | 24.9 | inv | rec | sca | L | part | sho | sa | NA | NA | NA | NA | NA | NA | NA | NA | NA |
| R19 | T.R.M. | 55 | 68 | 29 | 132.4 | dir | den | ste (D); sp (R) | D; R | cont (R); part (D) | sho (D); lon (R) | ab | ab | p/conc | p/conc | ab | 90 | 75 | 60 | >90 | >50% |
| R20 | Y.L.P. | 93.3 | 53.5 | 25.5 | 117.3 | dir | irr | ste (L); par (R) | R; L | cont | sho | sa (R); ab (L) | ab | ab | b/p | p/conc | >90 | 90 | 60 | 45 | <20% |
| R21 | Y.L.P. | 78.6 | 48.4 | 27 | 108.3 | dir | rec (R); irr (L); conv (D) | par (D); ste (R, L) | D; R; L | cont (L); part (R, D) | lon | sa | a | p/conc | b/p | p/conc | >90 | 55 | 60 | 60 | <20% |
| B1_1 | Y.L.P. | 78.8 | 66.9 | 29.5 | 152.3 | dir | conv | par (D); sca (L, R) | L; D; R | cont (D, R); part (L) | sho | low (R, D); sa (L) | ab | p/conc | p/conv | p/conv | >90 | 45 | 35 | 45 | 20% |
| B1_2 | Y.L.P. | 79 | 65 | 19.2 | 109.9 | dir (R); cro (L) | rec | sca (R); ste (L) | L; R | cont (R); part (L) | sho | ab (L); low (R) | ab | conv/p | b/p | ab | >90 | 60 | 35 | 85 | <20% |
| B1_3 | Y.L.P. | 83 | 73.5 | 22 | 122.2 | dir | conv | sca | L; D; R | cont | lon | sa | ab | p/conv | p/conv | p/conv | >90 | 50 | 40 | 45 | 20–50% |
| B1_4 | Y.L.P. | 60.8 | 46.2 | 19.5 | 52.4 | dir | conve (D); rec (L) | ste | D; L | cont (D); part (L) | sho | low | ab | b/p | b/p | b/p | >90 | 35 | 45 | 35 | 20% |
| B2_1 | Y.L.P. | 41.8 | 54.6 | 20 | 46.9 | dir (R); cro (P) | irr (R); rec (P) | ste | R; P | cont | sho | low | ab | b/p | b/p | ab | >90 | 50 | 40 | >90 | none |
| B2_2 | Y.L.P. | 51.6 | 50.4 | 21 | 65 | dir | conv | ste | L; D; R | cont (D); part (L, R) | sho | sa | ab | p/conv | p/conv | p/conv | >90 | 70 | 75 | 55 | <20% |
| B2_3 | Y.L.P. | 54.9 | 56.25 | 21.3 | 48.6 | dir | conv | ste | D; R | part | sho | sa | ab | p/conv | b/p | ab | >90 | 65 | 45 | 90 | 20–50% |
| B2_4 | Y.L.P. | 68.3 | 24.6 | 18.6 | 22 | dir | poin (D); conv (R) | sca | D; R | cont | sho | low | ab | b/conv | conv/p | b/p | >90 | 75 | 50 | 35 | none |
| B3_1 | Y.L.P. | 98.3 | 57 | 22.5 | 112.9 | dir (L); inv (P) | rec | ste | L; P | cont | sho | low (L); ab (P) | ab | b/conv | b/p | b/p | 90 | 90 | 35 | 30 | 50% |
| B3_2 | Y.L.P. | 10.4 | 38 | 14.7 | 57.7 | dir | conv | ste | R | part | sho | low | ab | b/conv | b/p | b/p | 90 | 55 | 25 | 55 | <20% |
| B3_3 | Y.L.P. | 81.8 | 52.8 | 22.6 | 79.9 | dir | conv | ste | D | part | sho | sa | ab | p/conc | b/p | p/conv | 90 | 50 | 55 | 60 | >50% |

(*Continued*)

**Table 4.** (Continued)

| Tool ID | Knapper | Max. length (mm) | Max. Width (mm) | Thickness (mm) | Weight (g) | Position | Delineation | Morphology | Localisation | Distribution | Extent | Angle | Proximal Cross-Section | Distal Cross-Section | Right Cross-Section | Left Cross-Section | Proximal Angle | Distal Angle | Right Angle | Left Angle | Cortex |
|---|---|---|---|---|---|---|---|---|---|---|---|---|---|---|---|---|---|---|---|---|---|
| B3_4 | Y.L.P. | 77.2 | 59.4 | 22.6 | 102.9 | dir | conv | lon; ste | L; D; R | cont (L); part (D, R) | sho | sa | ab | p/conv | p/conv | b/p | >90 | 55 | 40 | 40 | 20% |
| B3_5 | Y.L.P. | 86.4 | 59.9 | 28 | 161.7 | dir | conv | sca | L; D; R | part (L); cont (D, R) | sho | low | ab | b/p | b/p | b/p | >90 | 30 | 40 | 40 | >50% |
| B4_1 | Y.L.P. | 72 | 70.2 | 24.6 | 141.7 | dir | conv | sca | D; R; L | part (L); cont (D); part (R) | sho | low | ab | p/conc | p/conv | p/conc | >90 | 50 | 60 | 75 | <20% |
| B4_2 | Y.L.P. | 60.2 | 55.3 | 28 | 89.3 | dir | rec L); conv (R, D) | ste (L, D); sp (R) | D; L; R | cont | sho (L); lon (D, R) | ab (D); sa (R); low (L) | ab | p/conv | b/p | b/p | >90 | 70 | 55 | 45 | <20% |
| B4_3 | Y.L.P. | 59.6 | 38 | 18.8 | 38.8 | dir | conv | par | D | part | sho | sa | ab | p/conv | p/conv | b/p | >90 | 50 | 60 | 35 | <20% |
| B4_4 | Y.L.P. | 72.6 | 64.9 | 32.5 | 159.2 | dir | conv | ste | D | cont | lon | sa | ab | p/conv | ab | p/conc | >90 | 60 | 90 | 45 | 20% |

Abbreviations: 'dir' = 'direct', 'inv' = 'inverse', 'bif' = 'bifacial', 'cro' = 'crossed', 'rec' = 'rectilinear', 'conv' = 'convex', 'irr' = 'irregular', 'den' = 'denticulated', 'poin' = 'pointed', 'ste' = 'stepped', 'sp' = 'sub-parallel', 'lon' = 'long', 'sca' = 'scaled', 'par' = 'parallel', 'cont' = 'continuous', 'part' = 'partial', 'sho' = 'short', 'lon' = 'long', 'inva' = 'invasive', 'ab' = 'abrupt', 'sa' = 'semi-abrupt', 'p/conv' = 'plan-convex', 'b/conv' = 'bi-convex', 'b/p' = 'bi-plan', 'conv/p' = 'convex-plan', 'p/conc' = 'plan-concave', 'b/conc' = 'bi-concave'. Letters in brackets identify the lithic edge: 'R' = 'right', 'L' = "left', 'D' = 'distal', 'P' = 'proximal'.

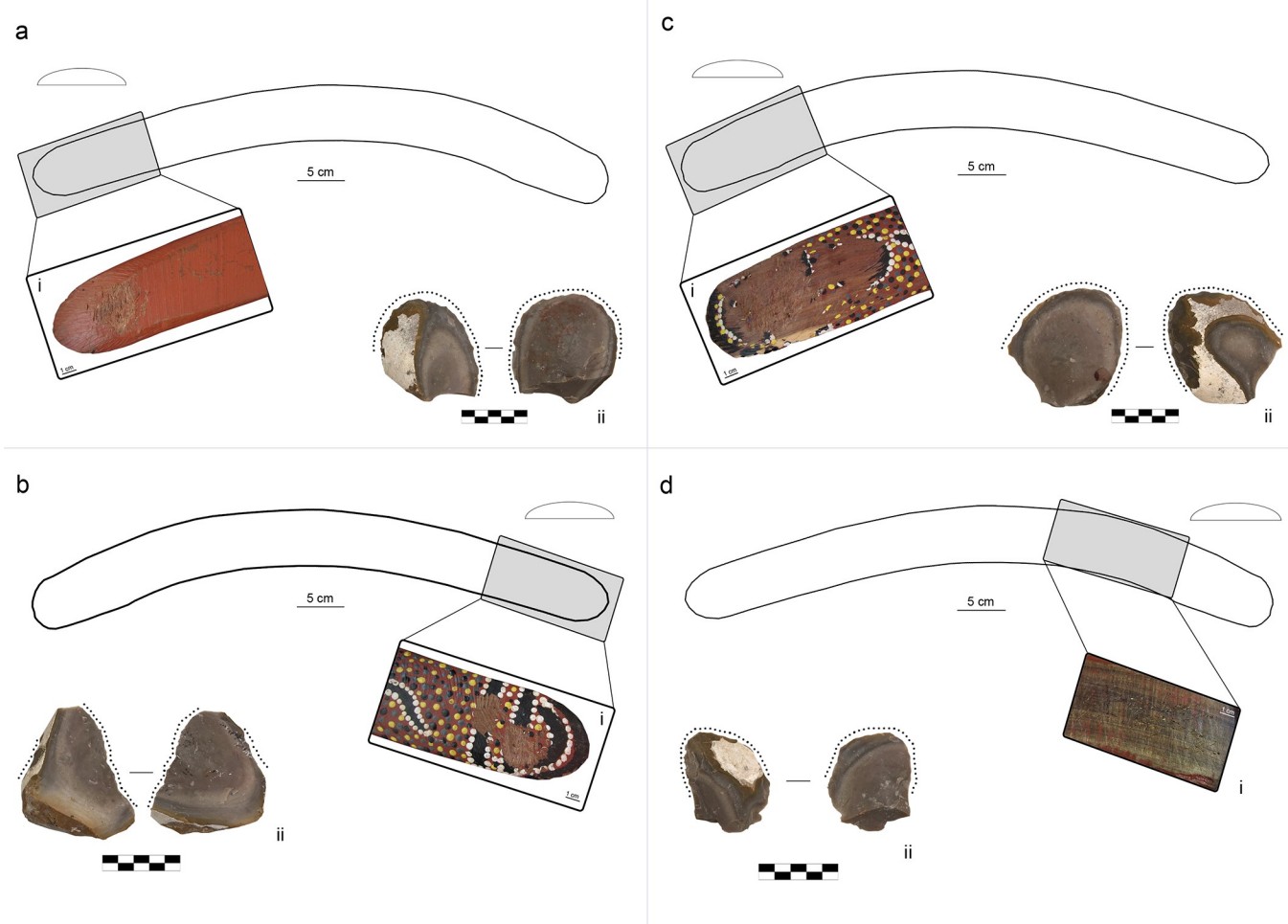

**Fig 2. Results of retouching session 2 on the boomerang B1 (plano-convex cross-section and location of use areas).** (a) location of use area B1_1, (i) use area B1_1 (1x), (ii) retouched flake F56; (b) location of use area B1_2, (i) use area B1_2 (1x), (ii) retouched flake F59. (c) location of use area B1_3, (i) use area B1_3 (1x), (ii) retouched flake F51; (d) location of use area B1_4, (i) use area B1_4 (1x), (ii) retouched flake F90.

boomerang was covered in paint by the original artist, and this paint was scraped away with sandpaper before the session but only at use area 3. The flake retouched with use area 1 shows residues of paint on its ventral surface, whose inclination follows the direction of retouch (Fig 1C). Use area 4 is located between the elbow and the tip (Fig 2D). Early on in the experiments on this portion of the boomerang's surface, the previously applied movement was deemed unsuitable because it injured the operator's thumb which used to stabilise the lithic flake. Therefore, the operator changed the retouching movement, still following an elliptical trajectory but aiming for a contact point that was more parallel contact to the lithic edge. The knapper, YLP, reported that this allowed for a more comfortable and balanced grasp of the boomerang. The flake retouched with the use area 4 (F90) fully meets the operator's technical objectives.

B2 was exploited in four portions of its surface, all near the tips (Fig 3). The number of blows is more variable, from 26 to 129 blows, and each use area was used for an average of two minutes. During the exploitation of use area 2 (Fig 3B), the retouched flake (F71) broke after 129 blows. According to the operator's technical objectives, it is considered successfully retouched. Finally, the exploitation of use area 4 (Fig 3D) ended after 26 blows because the retouched flake (F83) did not require an intense retouch activity.

B3 is the only boomerang exploited in five portions of its surface, all located near the tips, except for the use area 3, located on the central portion of the arm (Fig 4). Like in the case of B1_4, the operator had to aim for a parallel contact point when using the portion of the boomerang's surface between the elbow and the tip. Each retouch activity lasted for an average of two minutes, and the number of blows varies from 50 to 145.

B4 was exploited in four portions of its surface, all in proximity to the tips (Fig 5). The average duration of each retouch activity is two minutes, with a minimum of 6 and a maximum of 80 blows. During the exploitation of use area 3 (Fig 5C), the activity was interrupted after six blows because the flake (F81) was deemed unsuitable for retouching as the working edge was too obtuse. We took the methodological choice of not starting a new retouching activity involving the same use area, and we decided to consider B4_3 as a "short retouching", potentially associable with a resharpening activity.

## Retouch-induced use-wear analysis: Comparing bone and boomerang retouchers

The morphological categories of retouch-induced impact traces typically present on Palaeolithic bone retouchers were identified on the experimental boomerangs. Moreover, the 3D

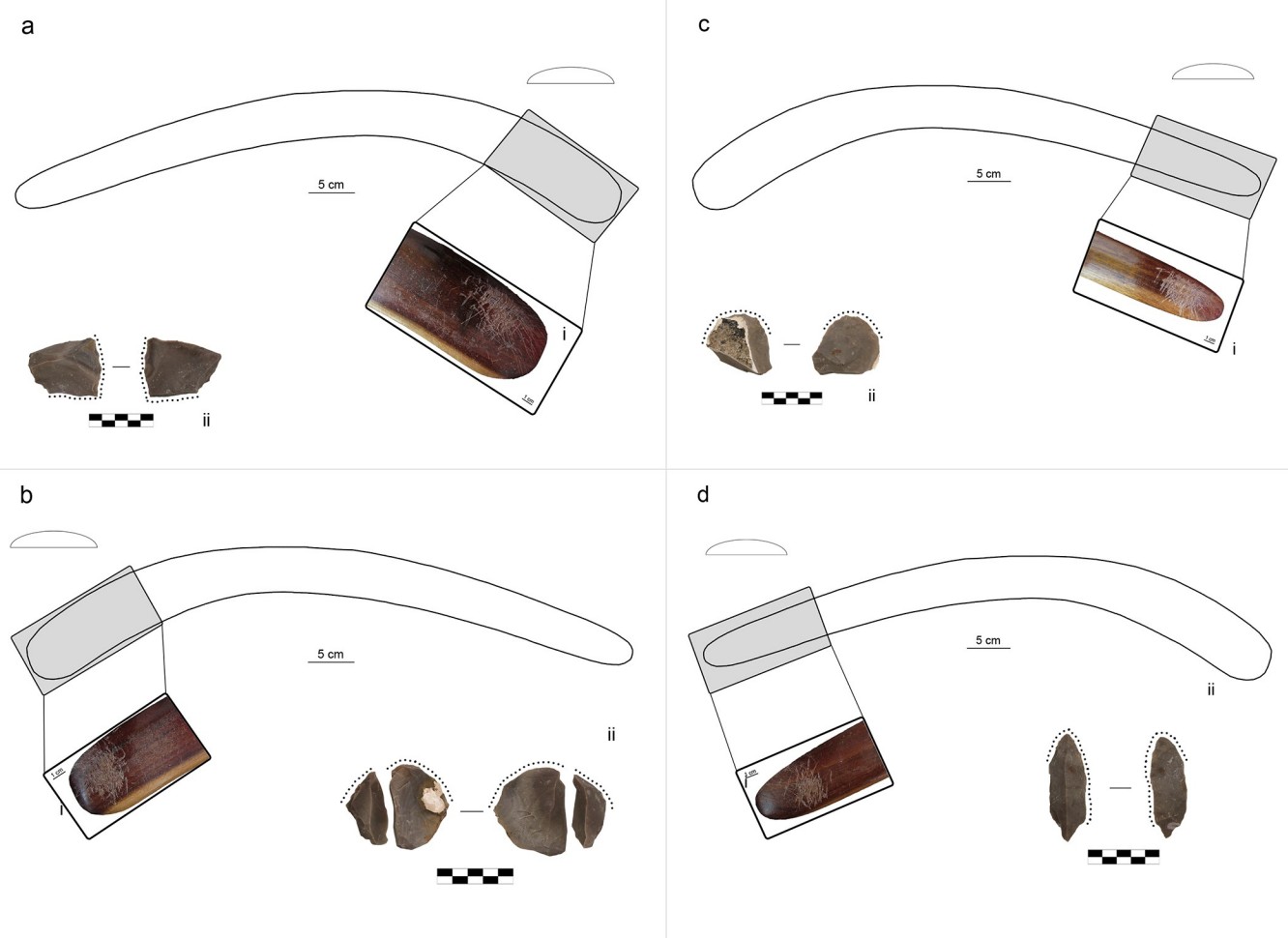

**Fig 3. Results of retouching session 2 on the boomerang B2 (plano-convex cross-section and location of use areas).** (a) location of use area B2_1, (i) use area B2_1 (1x), (ii) retouched flake F100; (b) location of use area B2_2, (i) use area B2_2 (1x), (ii) retouched flake F71. (c) location of use area B2_3, (i) use area B2_3 (1x), (ii) retouched flake F17; (d) location of use area B2_4, (i) use area B2_4 (1x), (ii) retouched flake F83.

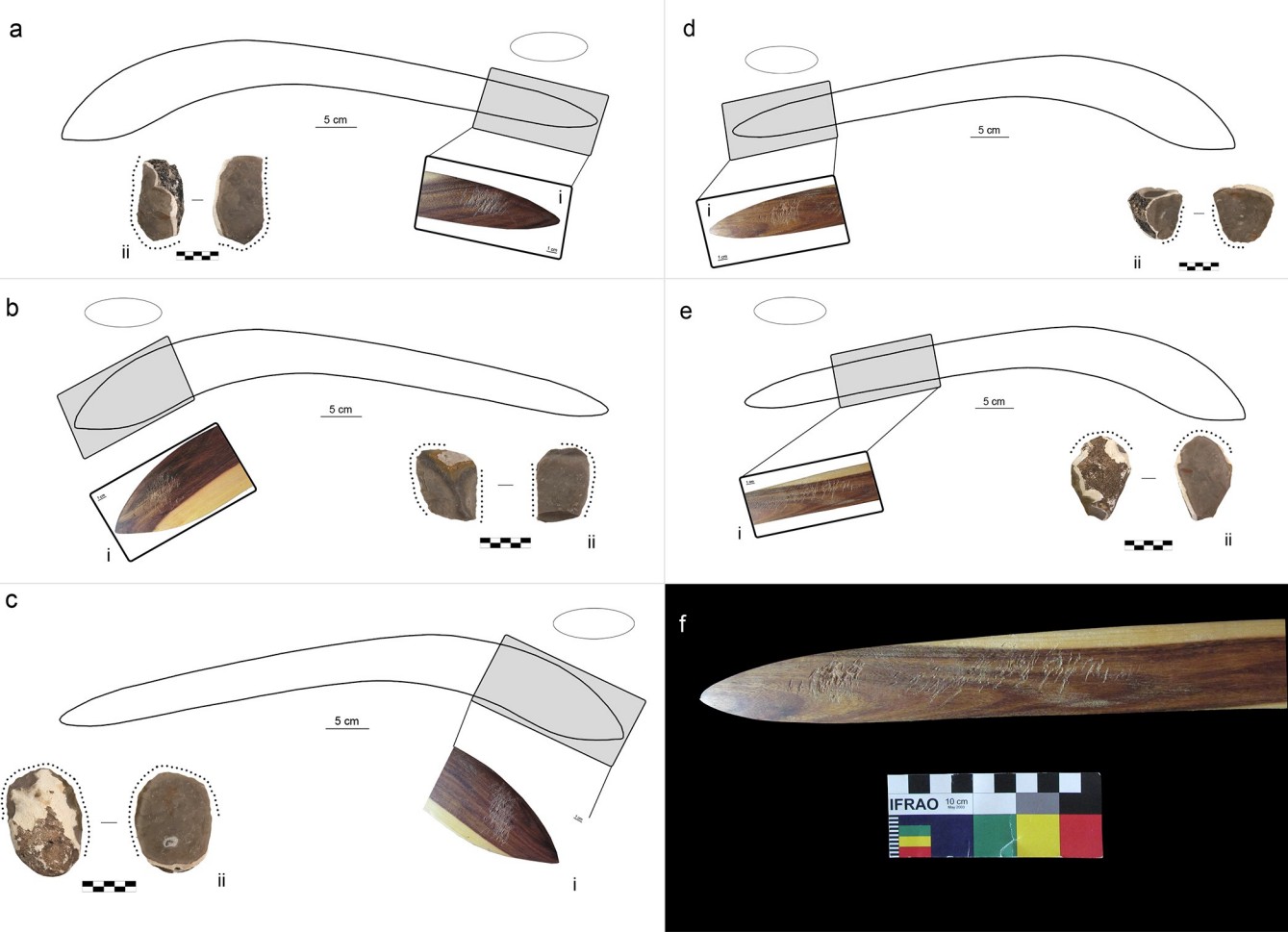

**Fig 4. Results of retouching session 2 on the boomerang B3 (bi-convex cross-section and location of use areas).** (a) location of use area B3_1, (i) use area B3_1 (1x), (ii) retouched flake F36; (b) location of use area B3_4, (i) use area B3_4 (1x), (ii) retouched flake F76; (c) location of use area B3_5, (i) use area B3_5 (1x), (ii) retouched flake F40. (d) location of use area B3_2, (i) use area B3_2 (1x), (ii) retouched flake F73; (e) location of use area B3_3, (i) use area B3_3 (1x), (ii) retouched flake F03; (f) adjacent use areas B3_2 and B3_3.

analysis of single impact traces revealed similar features between osseous and wooden samples of retouchers.

## Metric data on use areas

Some of the 14 bone retouchers used during session 1 show more than one use area; therefore, a total of 22 use areas were investigated on bone retouchers. Three of the four boomerangs used in retouching session 2 have four use areas each, and one boomerang was exploited in five portions of its surface; therefore, a total of 17 use areas on boomerangs were analysed (Table 3).

The length of the use areas on bone retouchers varies from 7.86 to 52.87 mm (average = 26.97 mm; $\sigma$ = 11.79 mm), whereas on boomerangs, the length of the use areas varies between 27.6 mm and 97.2 mm (average = 45.5 mm; $\sigma$ = 21.06). The width of use areas on bone retouchers varies from 6.12 to 30 mm (average = 20.94; $\sigma$ = 6.53) and on boomerangs from 19.2 mm to 43.7 mm (average = 30.1 mm; $\sigma$ = 7.68). The surface of the use areas on bone retouchers varies from 114.2 mm$^2$ to 771.4 mm$^2$ (average = 320.31 mm$^2$; $\sigma$ = 169.25), whereas on boomerangs it varies from 319.2 mm$^2$ to 2192.9 mm$^2$ (average = 818.05 mm$^2$; $\sigma$ = 452.45).

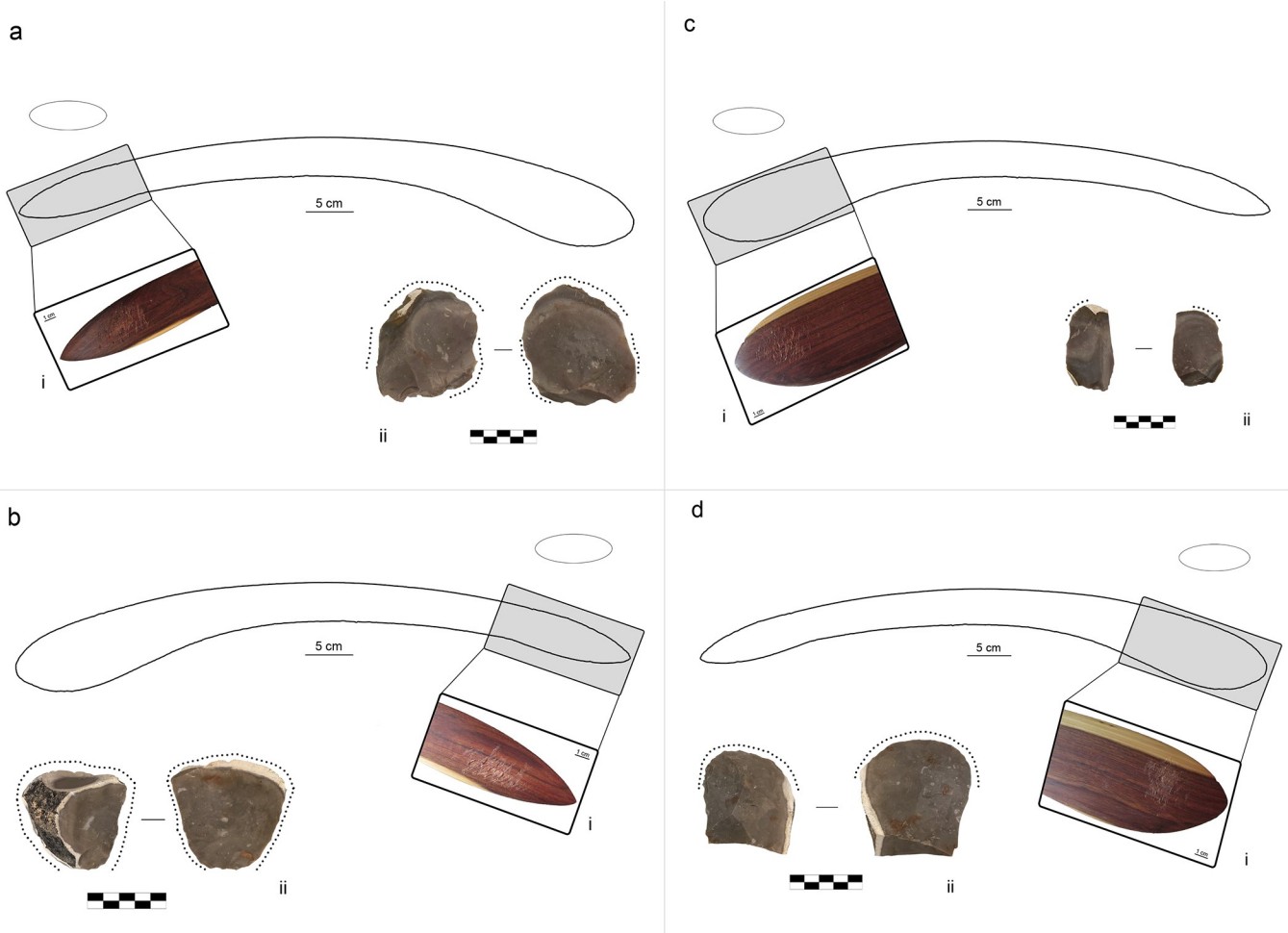

**Fig 5. Results of retouching session 2 on the boomerang B4 (bi-convex cross-section and location of use areas).** (a) location of use area B4_1 (i) use area B4_1 (1x), (ii) retouched flake F60; (b) location of use area B4_2(i) use area B4_2 (1x), (ii) retouched flake F35. (c) location of use area B4_3, (i) use area B4_3 (1x), (ii) retouched flake F81; (d) location of use area B4_4(i) use area B4_4 (1x), (ii) retouched flake F41.

Finally, the perimeter of the areas on bone retouchers varies from 22.9 to 201.9 mm (average = 97.8 mm; σ = 38.16) and from 114.9 to 435.4 mm on boomerangs (average = 255.1 mm; σ = 80.49). These data are summarised in S6A Fig. The measurements of the use areas are, in general, more consistent on boomerangs and more variable on bone retouchers.

Moreover, it appears that the use areas are bigger on boomerangs than on bone retouchers. This difference could result from the significant difference in the size of the two types of tools, i.e., boomerangs have a greater surface than bone retouchers, and therefore the use areas are likely to be greater as well. Such difference could be only partially appreciated in lengths and widths, but it looks much more evident when comparing surfaces and perimeters of the use areas, which are almost double the size of boomerangs. Nevertheless, the influence of deep structural differences between osseous and wooden material cannot be excluded. Further mechanical and physicl tests should be performed to clarify this issue.

## Morphological analysis of the impact traces

The analysis of the intensity of the retouch on bone retouchers revealed that most of the use areas are concentrated (36%; N = 8) and dispersed (32%; N = 7), followed by 27% (N = 6)

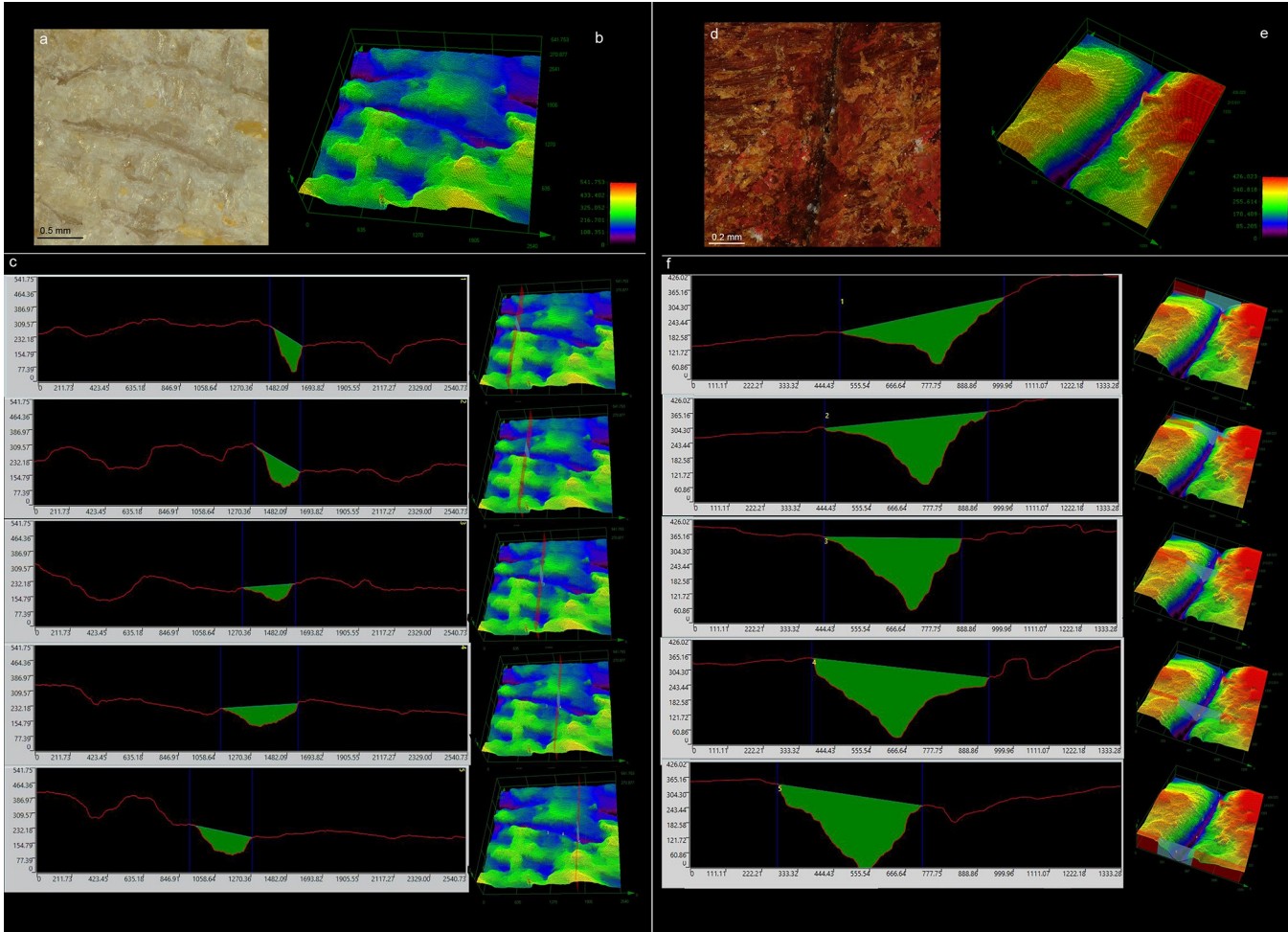

**Fig 6. 3D analysis of a single linear impression on bone retoucher (left) and boomerang (right).** (a) colour 2D image of a linear impression on bone retouchers (10x); (b) image of the same stigma with 3D height data; (c) V-shaped profile measurements and cross-section internal surface measured in different portions of the linear impression on bone retoucher; the deep penetration of the use mark in the bone surface can be observed. (d) colour 2D image of a linear impression on boomerang (10x); (e) image of the same stigma with 3D height data; (f) V-shaped profile measurements and cross-section internal surface measured in different portions of the linear impression on boomerang; the deep penetration of the use mark in the wooden surface can be observed.

being superposed distribution and 5% (N = 1) are isolated areas (S6BA Fig). On boomerangs, the majority of the use areas are concentrated (65%; N = 11), followed by dispersed (18%; N = 3) and superposed (18%; N = 3). No isolated use areas were identified on the utilised boomerangs (S6BB Fig). Although this result might be interpreted as a difference in intensity and distribution of use-wear between bone and boomerang retouchers, it is worth noting that the retouching session 1 involved two knappers (T.R.M. and Y.L.P.) whereas session 2 involved only one (Y.L.P.). As shown in S6BC Fig, when the two operators are involved, the retouch intensity seems to be related to the knapper's personal techniques.

The analysis of the impact traces (Table 3) shows that the four morphological categories usually identified on bone retouchers are also present in the use areas of the boomerangs. The 3D analysis allowed us to appreciate similarities with a high level of detail.

Linear impressions on bone retouchers (Fig 6A and 6B) are characterised by an elongated shape and an asymmetrical V-shaped profile, deeply penetrating the osseous surface (Fig 6C). The same diagnostic features are present on the surface of the boomerang: deep, elongated use

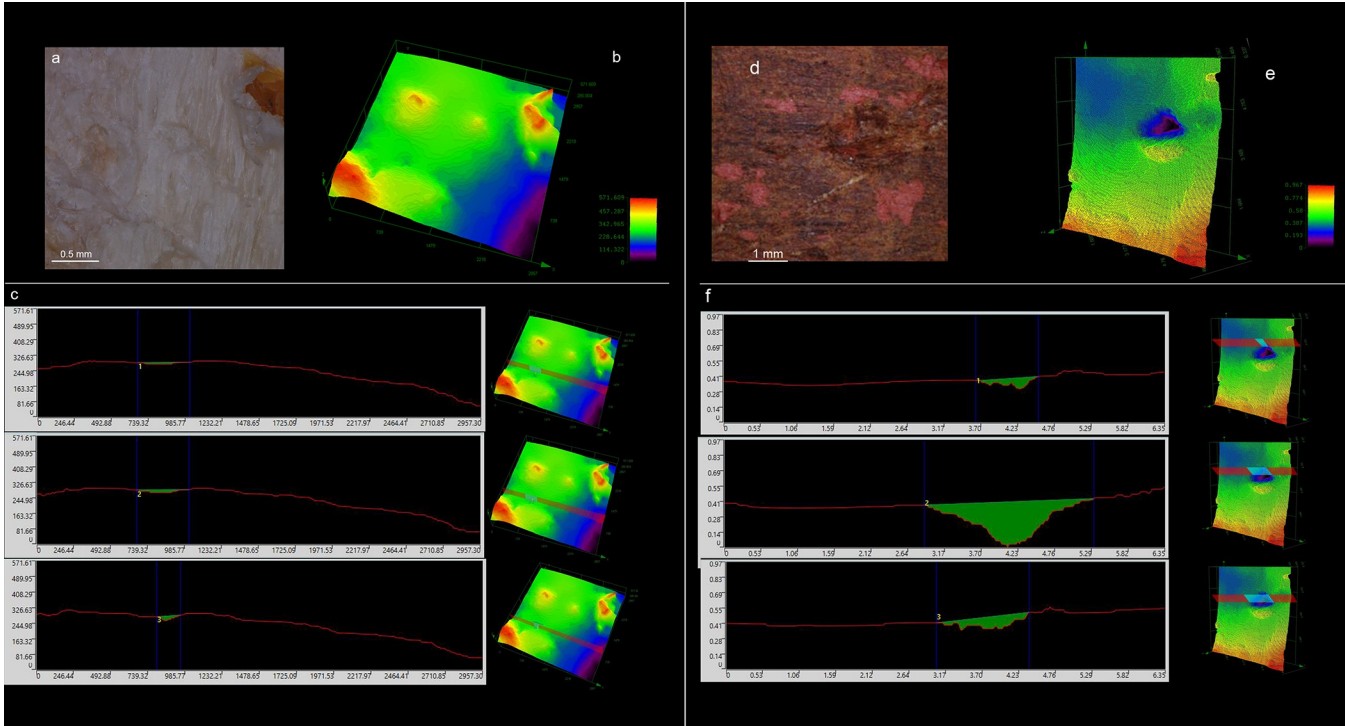

**Fig 7.** 3D analysis of a single punctiform impression on bone retoucher (left) and boomerang (right). (a) colour 2D image of a punctiform impression on bone retoucher (10x); (b) image of the same stigma with 3D height data; (c) U-shaped profile measurements and cross-section internal surface measured in different portions of the punctiform impression on bone retoucher; we can observe how the punctiform mark does not penetrate deeply in the bone surface. (d) colour 2D image of a punctiform impression on boomerang (3x); (b) image of the same stigma with 3D height data; (c) U-shaped profile measurements and cross-section internal surface measured in different portions of the punctiform impression on boomerang; we can observe how the punctiform mark penetrates deeply in the wooden surface, but it remains shallower than the linear impression.

marks have been identified as linear impressions (Fig 6D and 6E) characterised by an asymmetrical V-shaped profile (Fig 6f). Punctiform impressions are defined as shallow, round depressions with a generally U-shaped cross-section, as they appear on our sample of experimental bone retouchers (Fig 7A–7C). These use marks are also present on boomerang use areas (Fig 7D–7F); most are shallow, although some can go deeper into the wooden surface.

The results of the traceological analysis revealed that linear impressions are predominant on both bone retouchers and boomerangs—respectively, 88% and 78.9% of the total identified impact traces. Punctiform impressions represent 11% of the identified impact traces on bone retouchers and 20.47% on boomerangs. However different in frequency (Table 3; S6C Fig), linear and punctiform impressions follow a similar distribution pattern on boomerangs and bone retouchers (S6D Fig).

The third identified category is notches (Fig 8); they represent 0.8% of the total use-wear on bone retouchers and 0.27% on boomerangs. Bone retouchers show a chaotic distribution of notches penetrating the bone surface in depth (Fig 8A–8C). Notches are also widely present on boomerangs (Fig 8D–8F). This type of use-wear exhibits an irregular profile, resulting from the repeated superposition of various impact traces. Indeed, the 3D analysis reveals that notches are generated from a superposition of several linear and punctiform impressions, interacting and obliterating each other (Fig 8C–8F).

Previous experimental studies on the use of bone retouchers proved that the association between the number of blows and impact traces is highly variable. Such proportion never reflects perfect equivalency (i.e., 1 blow = 1 stigma), making the retouch activity susceptible to

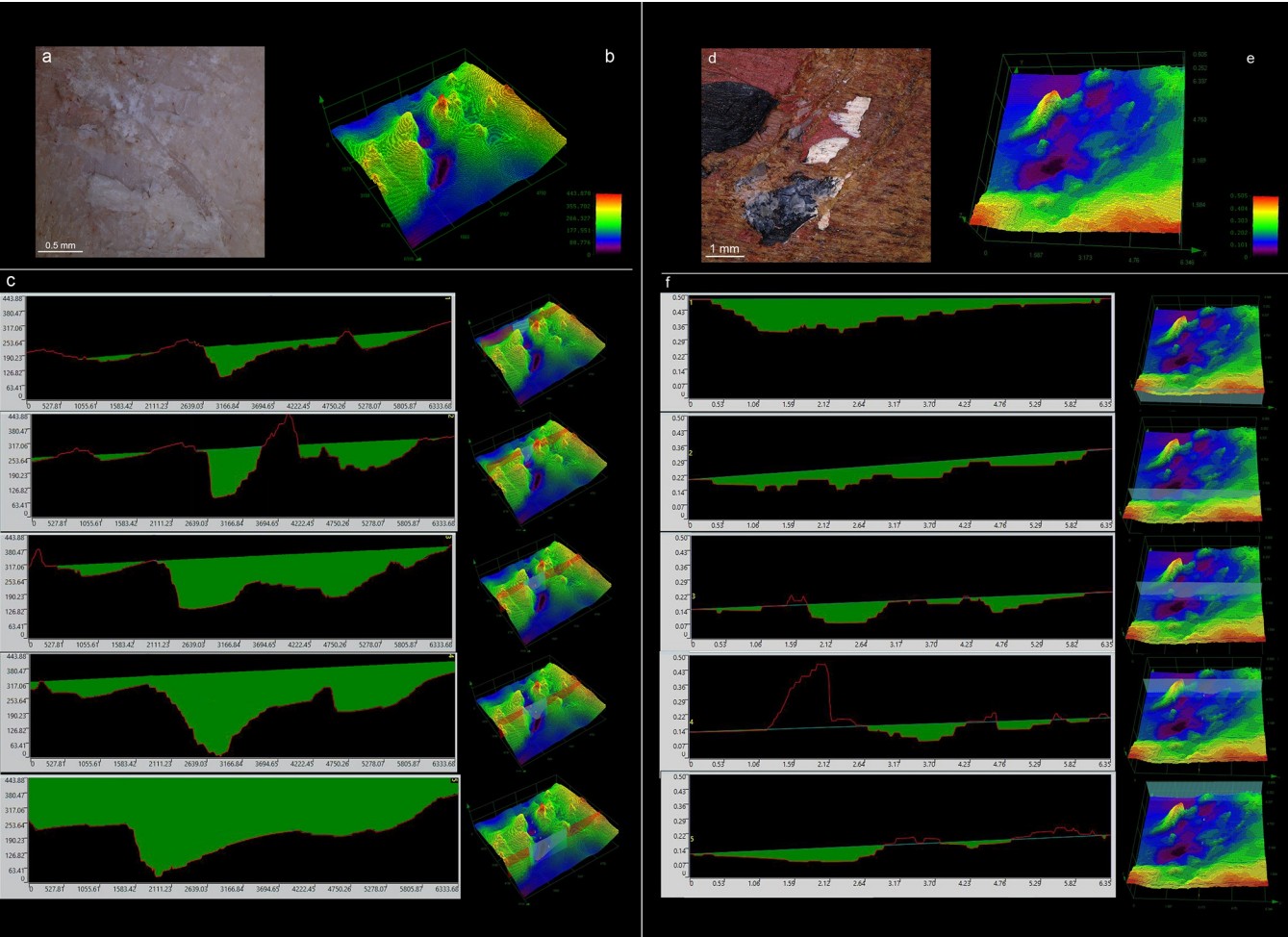

**Fig 8.** 3D analysis of a single notch on bone retoucher (left) and boomerang (right). (a) colour 2D image of the notch on bone retoucher (10x); (b) image of the same stigma with 3D height data; (c) profile measurements and cross-section internal surface measured in different portions of the notch on bone retoucher; the analysis shows that the notch is composed of the superposition of linear and punctiform impressions, penetrating deeply in the bone surface. (d) Colour 2D image of the notch on boomerang (3x); (b) image of the same stigma with 3D height data; (c) profile measurements and cross-section internal surface measured in different portions of the notch on boomerang; the analysis shows that the notch is composed of the superposition of linear and punctiform impressions, penetrating very deeply in the wooden surface.

several variables. However, our results reveal a correlation between the number of blows and the extension of notches. For instance, B3_4 shows one single, extended notch, and it is the use area with the highest number of blows (145); in this use area, linear marks are present in an average number. In contrast, in use areas formed by a smaller number of blows (e.g., B4_3), notches are absent or less extended, and linear impressions and striations are common (Table 3). Therefore, it is possible to test the existence of a negative proportional correlation between linear impressions and notches, dictated by the retouch intensity (S6E Fig).

The last morphological category associated with retouch activity consists of striations. They are present in 0.6% of bone retoucher use areas and 0.38% of boomerang use areas. These marks are short, shallow, and often grouped and parallel to each other. They appear very clear on the wooden surface of boomerangs (Fig 9A).

Finally, scraping marks have been identified in large numbers on bone retouchers and boomerangs. They are associated with 12 use areas (55%) distributed on ten retouchers and are present on 88% (N = 15) of the use areas on boomerangs. Scraping marks are shallow and

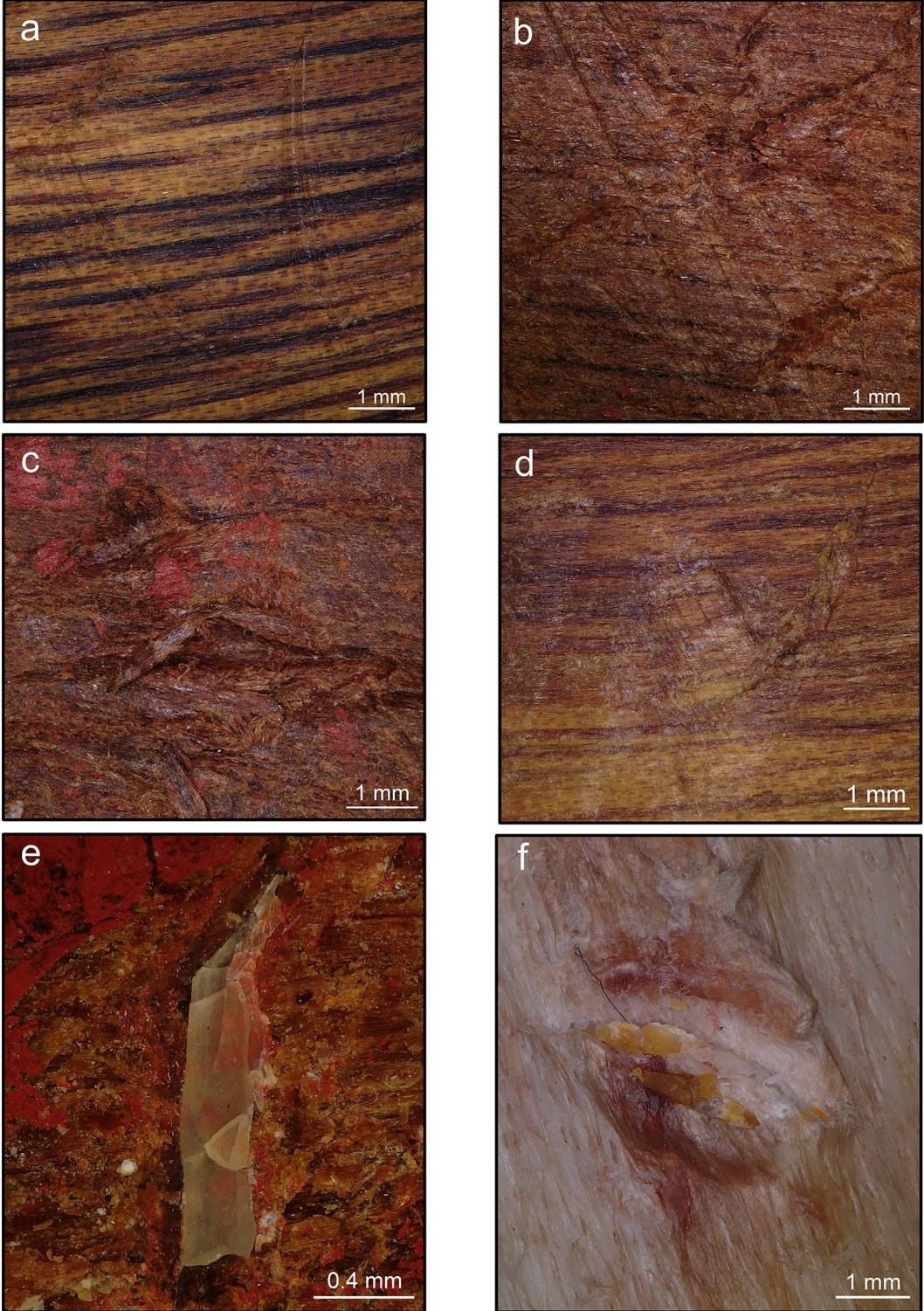

**Fig 9. Images of other marks identified within the use areas on boomerangs.** (a) striations associated with punctiform and linear impressions (3x); (b) scraping marks interacting and intersecting with linear and punctiform impressions (3x); (c) peeling-like use-wear (3x); (d) tool-edge scratches interacting and intersecting with linear impressions (3x); (e) flint chip embedded in linear mark on boomerang surface (10x); (f) flint chip embedded in linear marks on bone retoucher surface (R6_2) (3x).

long, and they often interact with other impact taces; they cover a greater portion of the use area than other marks (Fig 9B).

These experiments also identified certain marks on the boomerangs' use areas which are rarely identified on bone retouchers. Some of these marks appear as small detachments of a thin layer of wooden surface in correspondence with linear impact traces (B3_3; Fig 9C); their features resemble the "peeling" marks observed on broken bones [41] or weathered bones used in knapping activities [27, 42, 43]. Furthermore, some short, shallow, and parallel marks were identified. They start from the edges of the linear impressions and extend perpendicularly. We defined them as 'tool-edge scratches' [44] (Fig 9D). Finally, a significant number of stone micro-flakes (i.e., only visible under magnification) were embedded in the impact traces, in both linear and punctiform impressions (Figs 9E and 10);

## Metric analysis of the impact traces

We carried out a metric analysis of the two most commonly identified morphological categories—linear and punctiform impressions—to enhance the comparison between use areas on bone retouchers and boomerangs. The length was measured for linear impressions and surface and perimeter for punctiform impressions. As shown in Table 3, a difference in frequency exists between impact traces identified on bone retouchers and boomerangs, which compromises the data visualisation (S6F Fig). To address this issue, we generated a purely random selection of impact traces measurements recorded from boomerangs (N = 256, i.e., the number of observations registered in the bone retouchers database). The resulting comparison suggests a general trend of bigger imperssions on boomerangs than on bone retouchers (S6G Fig; see also S6H–S6J Fig), consistently with the metric data of use areas. Punctiform impressions, in particular, seem to be influenced by the size of the retouching tool, since boomerangs show a more significant variability of the surface covered by the pits. In contrast, this surface is smaller and less variable on bone retouchers.

## Embedded micro-flakes

Finally, the observation of use areas at high magnification and definition revealed the presence of several microscopic flint chips or splinters embedded in the retoucher surfaces. More than 60 micro-flakes were identified in one use area. These flakes were also identified on the experimental bone retouchers (Fig 9F). During retouching session 2, we observed a chip detached from the lithic edge after a few blows remained attached to the impact traces (Fig 1D). At the end of the retouching activity, the chip is no longer visible to the naked eye, but its remains can be found within the impact traces when magnified (Fig 10). It is possible to appreciate how the morphology of the chips and their fracture due to repeated percussion are responsible for creating linear or punctiform impressions. For instance, an elongated chip creates a linear impression (Fig 10A and 10b), whereas the morphology of the impact traces seems to change when the chip fractured by subsequent repeated percussion associated with the continuing retouch activity (Fig 10C–10F).

## Discussion

The multipurpose nature of boomerangs is a well-established concept in Australian Aboriginal Traditional knowledge, although it has received little direct investigation by archaeologists and ethnographic researchers [e.g., 5–9, 20]. In our recent work [5], we performed a systematic quantitative review of the literature available on the subject of "boomerangs". Our analysis showed how most previous boomerang-focused publications mainly consider the aerodynamic properties linked to the boomerang's infamous returning abilities. Technological and

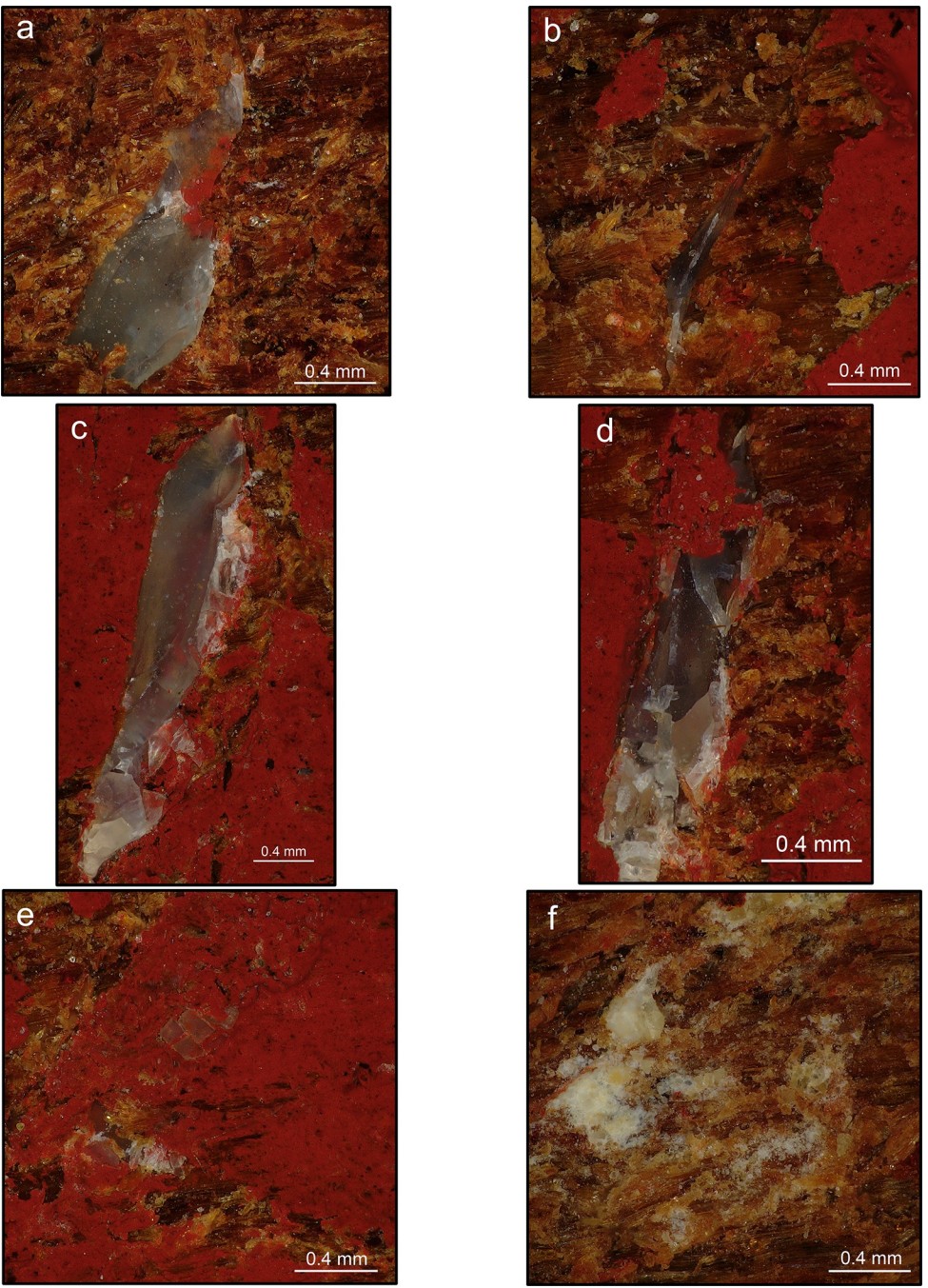

**Fig 10. Flint chips embedded within the use areas on boomerangs (10x).** (a) embedded flint chip mostly intact, showing paint residues owing to the use of a painted surface of the boomerang for retouching; (b) elongated flint chip creating a linear impression; (c) embedded flint chip fractured in its extremities following repeated percussion: punctiform impressions are more likely to be created as a consequence of these fractures; (d) embedded flint chip fractured in most of its surface due to repeated percussion: the micro-flakes resulting create punctiform impressions; (e) small, thin flint chip showing how each fracture of the chip itself from repeated percussion creates new impressions on the boomerang surface; (f) as a result of repeated percussion, several embedded flint chips are completely fractured leaving punctiform impressions and notches in the wooden surface.

functional aspects of non-returning boomerangs—including their use as retouchers—can be found in the form of incidental citations within broader descriptions of the Indigenous Australian lifestyle and daily activities (e.g., woodworking) [5]. Consequently, the remarkable variability of non-returning boomerangs is frequently overlooked.

During the second half of the 20th century, ethnographic reports on boomerangs are present in the literature, but they seem to stagnate in approaches and references belonging to the beginning of the century. Although their contribution to the current knowledge on boomerangs has to be recognised, it could also be argued that they are nowadays less suitable to contribute to a multidisciplinary and more scientific approach to ancient technology [5]. Suffice it to know that Davidson's work from 1936 [45] currently remains the main comprehensive study on boomerangs. Davidson himself concluded his contribution (p. 90) by wishing for an approach to boomerangs that was more technology-inclined than typological [5, 45].

Regarding recent contributions boomerangs have been rarely studied: this is a relevant issue, considering how much theories and techniques for the study of ancient technologies evolved in the last 30 years. Nevertheless, in this timeframe boomerangs have only been the object of sporadic classifications [7, 25, 46]; in other cases, they played a marginal role in valid experimental programs [20, 46]; finally, they were summoned when rare archaeological remains of boomerangs were discovered [15, 24]. This approach has overshadowed the role that these tools have played, and keep playing, in Australian Indigenous societies.

### The functionality of boomerangs in retouching lithic tools

From a purely functional standpoint, our experiment proved that boomerangs are effective tools for retouching lithic flake blanks, confirming literature evidence [5]. This conclusion was supported by the retouched lithic edges showing features consistent with the type of retouch sought-after by the operator (Table 3) and the collection of positive feedback from the knapper regarding the comfort in grasping and handling the boomerang during the retouch activity.

These technical objectives were achieved in both retouching sessions 1 and 2, demonstrating a similar functionality of boomerangs and bones in retouching activities. The movement applied to the successful use of boomerangs as retouching tools resulted comparable to the one observed in experiments with bone retouchers—that is, a percussion movement following an elliptical trajectory, with a tangential point of contact of the wooden surface against the lithic edge. However, a subtle difference between the two retouching tools could be found in the extension of the elliptical trajectory: the diameter of the ellipse drawn by the trajectory of the boomerangs seems to be slightly shorter than the one characterising the use of bone retouchers. Such difference is most likely due to the difference in sizes among the two types of tools: the boomerang, being larger and heavier than a bone blank, influenced the percussion movement, although it impacted neither the knapper's ability to perform the retouch nor the resulting use-wear (see below). According to the operator, the best position to use the boomerang as a retoucher is to align the concavity created by the elbow with the trunk of the operator's body: this position appears to be the most comfortable in grasping the boomerang with the right hand (Fig 1A and 1B) for a right-handed knapper. These observations can be compared with the study of the spatial distribution of use areas on asymmetrical boomerangs in museum collections [4: Fig 6A] (S5A Fig). The portions of the boomerangs showing a higher concentration of use areas—i.e., in the proximity of the tips—could be consistent with the knapper's choice of holding the boomerang in the most comfortable grasp. Among the same museum-curated samples, symmetrical boomerangs showed a more chaotic spatial distribution of the use areas, with most of them located on the central portion of the boomerang's arm [4: Fig 6B] (S5B Fig). To verify our hypothesis of a potential relationship between the retouching movement and the

spatial distribution of the use areas on the boomerang's surface, the operator attempted the retouch using a portion of the boomerang proximal to the elbow. It happened in two instances (B1_4 and B3_3). According to the operator, this technique appears to be less comfortable than a retouch performed using surfaces closer to the tips. The modification of the impact point—almost parallel to the lithic edge, rather than tangential (S5 File)—resulted in a functional retouch, but it generated more elongated use areas composed of mostly dispersed impact traces (Figs 2 and 4). However, these traceological observations are not compatible with what emerged from museum-curated symmetrical boomerangs [4].

In conclusion, our experiment revealed that both symmetrical and asymmetrical boomerangs are ultimately functional in retouching activities. The alignment of the boomerang's internal angle with the knapper's body may have greater importance than the morphology of the boomerang itself. When this condition was not met, the retouch was still successful, although the operator had to control the balance of the tool more carefully during the percussion movement. Therefore, the issue of the atypical spatial distribution of use areas on symmetrical boomerangs remains an open question. The reduced comfort in grasping could be among the reasons for the limited presence of symmetrical boomerangs bearing retouch-induced impact traces in the analysed museum-curated sample [4], although geographic and cultural factors must not be excluded.

## Traceological analysis and implications for archaeological studies

In analysing use areas on boomerangs, we recognised similarities in morphology and distribution of the use-wear with our experimental bone retouchers, concluding that the use marks on boomerangs can be defined as 'retouch-induced impact traces'. The typical V-shaped cross-section of linear impressions (Fig 6F) and the U-shaped, shallow depressions defined as punctiform impressions (Fig 7F) match the data present in several previous studies on bone retouchers; including both experimental and archaeological analyses of the cross-section of the impact traces and not only their morphology [27, 29, 30, 33, 36, 47–51, among others]. The frequency of linear and punctiform impressions on experimental boomerangs is comparable with our experimental bone retouchers (S6C and S6D Fig) and consistent with previous studies.

However, the main reasons behind the creation of linear rather than punctiform impressions still need to be investigated [see discussions in 31, 32]. On this topic, our microscopic analysis of use areas on boomerangs revealed a new element regarding the formation of impact traces, which appears to be related to the morphology of microscopic flint chips penetrating the working surface of the retouchers. Indeed, in several instances, the morphology of the stigma seems to be shaped by the flint pieces embedded in it: an elongated micro-flake leaves a linear impression on the wooden surface (Fig 10A, 10B), whereas the impacts of the retouching percussion repeatedly fracture the embedded chips and in turn contribute to the final shape of punctiform impressions and notches (Fig 10C–10F). Flint chips embedded in the impact traces have been observed in bone retouchers from experimental and, more rarely, Palaeolithic contexts [27, 29, 44]. Some micro-flakes were also present in our sample of experimental bone retouchers (Fig 9F).

Apart from the two main morphological categories (linear and punctiform impressions), other types of impact traces were also identified on boomerangs. A 3D analysis allowed us to view the topography of the notches (Fig 8F), and a cross-section analysis shows that the subsequent superposition of linear and punctiform impressions is responsible for the creation of the notches. Moreover, the experimental nature of our study addressed the issue of use areas composed of numerous notches as a result of an intense retouch activity. In the study of bone

retouchers, the intensity of use plays a crucial role in understanding the relationship that Palaeolithic hunter-gatherers had with the osseous raw materials used for making retouchers. This topic is often overlooked, and case studies comparing bone retouchers and lithic tools from the same context are rare. However, a pattern emerges when these aspects are compared, suggesting that lightly used bone retouchers are associated with assemblages containing a low number of retouched lithic tools [30, 31, 52]. In contrast, sites with numerous intensively utilised bone retouchers usually contain more retouched lithic flakes [31, 53]. Investigating new ways to reveal the relationship between retouched flakes and retouchers' traceology, therefore, deserves more thorough attention in archaeological studies. In this context, our results can prove helpful during the analysis of archaeological osseous or wooden retouchers since they could allow speculations on the degree of exploitation of organic raw materials by past societies.

Moreover, the traceological analysis revealed the presence of several scraping marks in association with use areas on our experimental boomerangs. In the technological analysis of bone retouchers, the origin of scrape marks has always been a debated topic–and whether these are the product of the retouch activity or the butchering processes, especially the removal of the periosteum [27, 32, 33, 43, 54, 55]. Our results contribute to this debate. In our experimental sample, scraping marks are present on more than half of the used bone retouchers made on various skeletal elements. Although the periosteum removal was carried out on some of the bones during the initial cleaning process, this action does not correlate to the presence of scraping marks. However, it appears that the knapper plays a relevant role in creating these marks: 70% of the use areas associated with one knapper present scraping marks, as opposed to the other knapper whose tools show the presence of scraping marks on 42% of the cases (Table 3; S6K Fig). Even more relevant is the observation of scraping on wooden retouching tools not subject to any butchering activity, which could be additional proof in favour of the retouch-related origin of scraping marks.

Furthermore, some infrequent use marks were identified in the use areas on the boomerangs. For instance, B3_3 presents some marks associable with 'peeling' traces [*sensu* 41] (Fig 9C). Considering the position of this use area (close to the elbow), this type of use mark could be associated with the unusual parallel trajectory of the retouching movement applied in this instance. Similar marks are also present in the use area B1_4 in association with linear impressions. It is interesting to note that peeling use-wear usually relates to bone fracturing actions [41]; to our knowledge, it is not commonly identified in the context of retouching use-wear, except for some instances in the use of weathered bones in knapping/retouching activities [27, 42, 43]. In addition, an uncommon category of use-wear associated with retouch activities was identified—the 'tool-edge scratches' [44, 56] (Fig 9D). They could be described as 'sliding marks', usually associated with linear impressions. Interestingly, these marks are present in the use area produced by a parallel trajectory, leading to the hypothesis that this use-wear could be connected to a different grasping technique of the retouching tool.

The technological and traceological analysis applied in the present work leads us to conclude that wooden and osseous materials react similarly to percussion stimuli. Such a conclusion is strengthened by our experimental results on boomerangs and the broad literature on bone retouchers in European contexts. However, a direct comparison with wooden retouchers is needed but lacking because of the long-term diagenetic processes in Palaeolithic sites [e.g., 57, 58]. For this reason, the experimental investigation of the functionality of hardwood boomerangs as retouchers could have great importance. Additionally, our study helps shed light on the role of wooden tools in Aboriginal Australian contexts.

On the one hand, wooden tools are rare in archaeological sites because of harsh Australian environmental conditions. On the other hand, non-archaeological wooden artefacts are

relatively numerous, but information regarding their functionality needs to be sifted out from the bias typical of the early European ethnographic reports [see discussions in 5]. Finally, percussion retouching techniques in Australian contexts have not been deeply investigated. Although the literature reveals some clues regarding the use of wooden tools as retouchers [4, 5], our new results bring empiric and experimental insights to this hypothesis.

## Future research

The present work represents the first experimental attempt at using hardwood boomerangs as retouchers, using a new methodology developed from several studies on bone retouchers. The main objective was to assess the functionality of boomerangs as retouchers. We attempted to minimise the variables—e.g., using only one type of lithic raw material, entrusting the retouching only to expert knappers, using similar types of hardwood to manufacture boomerangs, and using only two shapes of boomerangs. This approach allowed a clearer identification of which parameters are involved in the production of impact traces on the wooden surface and the relationship with the percussion movements characteristic of the retouching technique. This choice was also dictated by the scarcity of information on wooden retouchers and to what degree their response to percussion stimuli is comparable to that observed on osseous materials.

As the functionality of boomerangs in retouching activities has been experimentally established, the next step will be to expand the set of parameters of the experiment to fully understand the relationship between percussion movement, wooden surfaces and the lithic edge. For instance, the role the knapper had in the distribution of impact traces and scraping marks might deserve investigation. More operators with differing levels of expertise and experience in replicating various lithic technologies should be included in future experiments. Our study only included two experienced knappers so as not to compromise the traceological comparisons; however, the ability of the knapper and their learning processes should always be considered when investigating the human past. Although previous studies on bone retouchers have indicated that different lithic raw materials do not significantly influence the features of the impact traces, it would be interesting to test if this also applies to wooden retouchers or if perhaps a coarser-grained stone may impose some limitations to the retouching. Similarly, any functional implications of the use of different hardwoods, or even softwoods in the retouching process, are worth investigating. Finally, as recent studies [59, 60] proposed methods to distinguish various types of soft hammers through the study of lithic detachments and scars, it could be rather intriguing to include wooden retouchers in this discussion.

## Conclusion

The experimental program in this study closes a circle of related evidence regarding the use of hardwood boomerangs as retouchers. We experimentally demonstrate that boomerangs can successfully function as retouchers and that their use as such creates use-wear consistent with that observed on bone retouchers. This use-wear includes mostly linear and punctiform impressions, closely distributed in small portions of the surface of the boomerang; such marks have been observed previously on museum-curated boomerangs [4].

New information regarding the processes leading to the formation of impact traces was revealed with the analysis of high-resolution 3D scans. It sheds new light on the process of repeated impacts on lithic chips on the surface of the retouchers and the effect on the observed impressions.

Moreover, this work represents one of the first experimental studies to demonstrate that osseous and wooden percussor respond similarly to percussion stimuli in the context of

retouch activities. Several traceological similarities among the impact traces strongly support this conclusion. This evidence could be a valid reference in the study of archaeological assemblages containing possible wooden implements, allowing for speculations on their function through comparisons with osseous tools.

Finally, this study systematically investigates one of the many functional purposes of hardwood boomerangs and could lead the way to uncover further evidence for the polyhedric toolkit of Australian Indigenous cultures.

## Supporting information

**S1 Fig. A.** Knapping of experimental lithic flakes. (a) cobble_1 before knapping; (b) cobble_2 before knapping; (c) bifacial edge reduction; (d) sample of produced flake blanks; (e) exhausted core after knapping of cobble_1; (f) exhausted core after knapping of cobble_2. **B.** Types of cross-sections of functional edges of lithic tools. From [28]. **C.** Description of retouched edges. From [39].
(JPG)

**S2 Fig. Methods of measurement of the main items involved in the experimental protocol.** (a) measurements of lithic flakes according to their debitage axis; (b) measurement of bone retouchers; recording of 'C' and 'H' values follows Neruda and Lázničková-Galetová (2018); (c) measurement of symmetrical boomerangs; (d) measurement of asymmetrical boomerangs. Drawings by E. F. Martellotta.
(JPG)

**S3 Fig. A.** Measurement method applied to bones before breaking. Drawing by E. F. Martellotta. **B.** Tools used during the bone breaking session. (a, b) flint flakes occasionally used to remove periosteum; (c) hammerstone 2, i.e., H2; (d) hammerstone 1, i.e., H1; (e) anvil. **C.** Bone breaking session. (a) choice of the impact point; (b) incipient fracture; (c-d) the bone is completely fractured through flexion applied to the incipient fracture; (e) spiral fracture; (f) detachment of bone retouchers from fractured diaphysis; (g) obtained bone retouchers and discarded epiphyses.
(JPG)

**S4 Fig. Experimental boomerangs.** (a) B1; (b) B2; (c) B3; (d) B4.
(JPG)

**S5 Fig. Spatial distribution (ArcGIS 10.2) of use areas on boomerangs.** (a) asymmetrical shape; (b) symmetrical shape. From: Martellotta EF, Wilkins J, Brumm A, Langley MC. New data from old collections: Retouch-induced marks on Australian hardwood boomerangs. J Archaeol Sci Reports. 2021;37: 102967. doi:10.1016/j.jasrep.2021.102967.
(JPG)

**S6 Fig. A.** Metric data on use areas identified on boomerangs (top) and bone retouchers (bottom). Statistical significance of data—Length of use areas: R = 0.21; p = 0.43. Width of use areas: R = -0.24; p = 0.36. Surface of use areas: R = 0.12, p = 0.64. Perimeter of use areas: R = 0.4; p = 0.88. **B.** Representation of retouch intensity on boomerangs and retouchers. (a) retouch intensity of use areas on boomerangs; (b) retouch intensity of use areas on bone retouchers. (c) different intensity of retouch produced by the two knappers during retouching session 1; note that T.R.M. produced a greater variability of impact traces distribution than Y. L.P. **C.** Distribution of the two main categories of impact traces (linear and punctiform impressions) on boomerangs and retouchers. Numbers on the x axis indicate the frequency of observed impact traces in a single use area. 'NA' stands for 'use areas not showing the

impressions category'. **D.** Scatter plot of liner and punctiform impressions on boomerangs and retouchers. Correlation coefficient with Pearson's method and confidence of intervals (95%) applied. **E.** Scatter plot showing negative correlation between notches and linear impressions on boomerangs. Correlation coefficient with Pearson's method and confidence of intervals (95%) applied. **F.** Linear and punctiform impressions metric data: Comparison between boomerangs and bone retouchers. Note how the difference in frequencies makes a proper comparison difficult. **G.** Metric data for linear and punctiform impressions: Comparison between a sample of measurements form boomerangs and bone retouchers (N = 256). The sample of 256 measurements was randomly generated in the R software using the 'sample()' function. **H.** Scatter plot showing correlations among the length measurements for linear impressions on boomerangs and bone retouchers. Correlation coefficient with Pearson's method and confidence of intervals (95%) applied. **I.** Scatter plot showing correlations among the surface measurements for punctiform impressions on boomerangs and bone retouchers. Correlation coefficient with Pearson's method and confidence of intervals (95%) applied. **J.** Scatter plot showing correlations among the perimeter measurements for punctiform impressions on boomerangs and bone retouchers. Correlation coefficient with Pearson's method and confidence of intervals (95%) applied. **K.** Distribution of scraping marks on boomerangs and bone retouchers. (a) scraping marks on boomerangs; (b) scraping marks on bone retouchers. (c) differences in the distribution of scraping marks produced by the two knappers during retouching session 1; note that Y.L.P. produced a greater variability of scraping marks than T. R.M.
(PDF)

**S1 Table. Details of lithic flakes obtained during the knapping session.** Among them, 32 were selected to be retouched (see Table 3 within text). 'NA' stands for 'Not Applicable'.
(PDF)

**S2 Table. Information on bones used to manufacture bone retouchers.**
(PDF)

**S1 File. Description of breaking actions applied to bones to produce retouchers.** In each template, the first picture is a photo of bones before the breaking; the second is a 360˚ view of the bone, with indication of the impact points (in chronological order) and the portion used for retouchers (in red); the third picture is a photo of the bone blanks selected to be used as retouchers (photo and drawings by E. F. Martellotta). To facilitate the identification of the selected blanks, each bone was divided in 20 portions following: Romandini M. Analisi archeozoologica, tafonomica, paleontologica e spaziale dei livelli Uluzziani e tardo-Musteriani della Grotta di Fumane (VR). Variazioni e continuità strategico-comportamentali umane in Italia Nord Occidentale: i casi di Grotta del Col della Stria. Dipartimento di Biologia ed Evoluzione. Università degli Studi di Ferrara. 2012. Available: https://iris.unife.it/handle/11392/2389242#. XaaWxugzaUk.
(PDF)

**S2 File. Video of retouching session 1.** T.R.M. and Y.L.P. using bone retouchers.
(MP4)

**S3 File. Details of retouch session 1.** In each template, the first picture represents a 360˚ view of the bone retoucher and indication of the use area(s); the second picture is a photo of the retouched lithic flake. Retouchers described within the text (i.e., R6, R8, R12, R21) are excluded from this set (drawings and pictures by E. F. Martellotta).
(PDF)

**S4 File. Video of retouching session 2.** Y.L.P. using a boomerang as a retoucher.
(MP4)

**S5 File. Video of parallel movement applied during retouching session 2.** Y.L.P. using boomerang.
(MP4)

## Acknowledgments

The authors thank Dr Luc Doyon and an anonymous reviewer for their constructive suggestions. We wish to thank Tessa Dux and Dr Tim R. Maloney (Griffith University) for their assistance in the experiment. The authors are grateful to the Milan Dhiiyaan mob for sharing their Traditional knowledge and supplying *bubarra/garrbaa/biyarr* (boomerangs) representative of the cultural and spiritual beliefs of the Wailwaan and Yuin people.

## Author Contributions

**Conceptualization:** Eva Francesca Martellotta, Jayne Wilkins, Michelle C. Langley.

**Data curation:** Eva Francesca Martellotta.

**Formal analysis:** Eva Francesca Martellotta.

**Funding acquisition:** Eva Francesca Martellotta.

**Investigation:** Eva Francesca Martellotta, Yinika L. Perston.

**Methodology:** Eva Francesca Martellotta.

**Project administration:** Eva Francesca Martellotta.

**Resources:** Eva Francesca Martellotta, Paul Craft, Jayne Wilkins.

**Software:** Eva Francesca Martellotta.

**Supervision:** Eva Francesca Martellotta.

**Validation:** Eva Francesca Martellotta.

**Visualization:** Eva Francesca Martellotta.

**Writing – original draft:** Eva Francesca Martellotta.

**Writing – review & editing:** Eva Francesca Martellotta, Yinika L. Perston, Jayne Wilkins, Michelle C. Langley.

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
