## [Decision Letter · Decision Letter 0]

18 May 2022

PONE-D-22-05291Beyond the main function: An experimental study of the use of hardwood boomerangs in retouching activitiesPLOS ONE

Dear Dr. Martellotta,

Thank you for submitting your manuscript to PLOS ONE. After careful consideration, we feel that it has merit but does not fully meet PLOS ONE’s publication criteria as it currently stands. Therefore, we invite you to submit a revised version of the manuscript that addresses the points raised during the review process.

We look forward to receiving your revised manuscript.

Kind regards,

Enza Elena Spinapolice, Ph.D

Academic Editor

PLOS ONE

Journal Requirements:

Reviewers' comments:

Reviewer's Responses to Questions

**Comments to the Author**

1. Is the manuscript technically sound, and do the data support the conclusions?

Reviewer #1: Yes

Reviewer #2: Partly

2. Has the statistical analysis been performed appropriately and rigorously? 

Reviewer #1: Yes

Reviewer #2: No

3. Have the authors made all data underlying the findings in their manuscript fully available?

Reviewer #1: Yes

Reviewer #2: Yes

4. Is the manuscript presented in an intelligible fashion and written in standard English?

Reviewer #1: Yes

Reviewer #2: Yes

5. Review Comments to the Author

Reviewer #1: In their manuscript entitled "Beyond the main function: An experimental study of the use of hardwood boomerangs in retouching activities", Martellotta and colleagues investigate the potential of Aboriginal Australian boomerangs as implements to shape stone tools and compare the resulting use wear to that observed on bone retouchers which are well-documented in Palaeolithic and Neolithic contexts. To address this two-fold issue, they conceived and implemented a thorough experimental program, complemented with sound statistical analyses. I wish to emphasize here the diligent work produced by the authors and their apparent ethics in sharing ALL the data resulting from their experiments; the SI are very impressive. This will surely provide future researches on the subject with a wealth of data that can effectively be used in comparisons. Overall the manuscript is well-structured and is definitely pertinent for publication in PLOS ONE.

Below, I have only a few minor suggestions which would need to be addressed before publication. I am presenting it line by line.

Lines 113-114, 154-156 and 213-215: The equipment used for photography and video recording is repeated. Perhaps is can be avoided by adding a single general statement (at the beginning or the end of the Materials and Methods section).

Lines 127 and 145: The letter C is used twice for two different meanings: line 127 it refers to the circumference of the whole bone, line 145 it refers to the width of the arc of a bone fragments (according to Neruda, Lasznikova-Galetova, 2018). In addition, on line 145, H is not defined. They authors might want to re-word the lines 145 to 149 to make it more comprehensible.

Line 188: the knapper T.R.M. is not an author, and realized who it was only after reading the Acknowledgments. Perhaps spelling their name when introduced the first time would be appropriate.

Lines 258-259: This information was already presented in the Materials and Methods section. It s necessary to have it here?

Lines 287-317 and 328-452: I have two small issues with these sections. First, in selecting only 4 of 14 retouchers for their description, they only present data from one knapper Y.L.P. The authors may argue that this is to allow comparison between bone and boomerangs, but it masks, if any, the (qualitative) variability between knappers. Second, the descriptions are somewhat redundant. The authors may wish to streamline these sections so the reader focuses more on the ensuing comparisons, which are absolutely necessary to answer the research questions stated in the introduction.

Line 288: "found both areas sufficient", please be more explicit.

Line 310: "was exploited in one portion", please clarify, in a single session?.

Line 320: I think the first sentence can be removed as it is seldom pertinent here to say the boomerangs were experimentally used for the same function as bone retouchers...

Lines 466-476: ensure the proper math sign and unit conventions are used throughout this paragraph.

Lines 526-560: I have a difficulty to understand how the percentage were calculated. It would appear the authors report here the sum of one mark type divided by the total number of marks of all types. This is slightly misleading as one blow can produce multiple marks (e.g., R10_1 with 4 blows that produced 58 marks) and another can produce none (e.g., R10_2 with 46 blows and only 10 marks or R11_1 with 12 blows and only 5 marks).

Lines 555-556: This statement should be statistically tested rather than assumed, especially since the authors have all the data available.

Line 572: "are often interact with other", poor wording.

Lines 576-577 and 737-738: Similar "peeling" can occur when weathered bone is used in knapping activities (see Mallye et al., 2012, Doyon et al., 2018, 2019).

Lines 580-583: The authors may want to delete these was the information is presented in the section "Embedded micro-flakes".

Lines 591-594: It is not clear why this was not presented in the Materials and Methods. It is not clear either where the maximum of 256 measurements recorded on bone retouchers comes from. And, as a consequence, it is difficult to interpret the figures presented in S9 Figs. 5-6.

Line 627: please cite some references for the archaeological and ethnographic research.

Lines 632, 650, 655, 687, 755: The authors refer to discussion or figures from other publications. Reproduction of these figures (if rights allow it) and synthesis of the main points relavent to the discussion should be included in the text rather than simply referring to other works.

Lines 692-693: "repetitive impacts [...] repeatedly fracture" , the wording is a little off.

Line 707: "A pattern emerges when it these aspects are..." , wording issue.

Lines 709-710: it is not clear if the authors refer to assemblages containing few lithic tool types or few lithic retouched tool specimens.

Lines 716-730: Perhaps it would be pertinent to include the references from Jéquier et al 2018 and Doyon et al 2019 on this subject and see how their results compared to those presented in this study.

Lines 797-798: "demonstrate that osseous and wooden percussor tools have physical and mechanical properties in common" , although this is not completely wrong, the physical and mechanical properties were not tested per se. What the experiment shows is that hardwood tools can be used as retouchers just as bone, and that the resulting use wear is strongly similar.

Lines 816-952: Ensure the reference section matches the guidelines for authors.

Tab. 1: some formatting issues can be resolved easily, i.e., titles, numbers on two lines after the points, etc.). It was not clear, however, why the use areas were separated into distinct lines, especially since the information is expected to be the same. Rather than duplicate the lines, I would remove the information on use area name, AND add a column on the number of use area per retoucher. You might want to transfer the column (n. of use areas) from Tab. 3 to Tabs. 1 and 2 where it is more pertinent.

Tab. 2: you might want to remove the "_1" from the tool ID column.

Tab. 4: Formatting issues. Line order should be R1, R2, R3, R4, etc.

S7 Figs. 1-4 (panel lithic tool): it would be interesting to highlight (perhaps only with a dash line) the location of the retouch along the edges.

S9 Fig. 1: The Y-axes are not the same for boomerangs and bone retouchers. Perhaps, the two tool types can be compared side-by-side for each metric.

S9 Figs. 3, 4: There are some discrepancies between the data shown in the figures and those presented in Tab. 3. In Tab. 3, the maximum linear impressions is 559 and 432 for boomerangs and retouchers respectively. The maximum number of punctiform impressions are 209 and 69. The title of S9 Fig. 4 should state "scatterplot" rather than "statistical distribution". The % of the confidence of intervals should be indicated.

S9 Fig. 5: the unit for perimeter should be mm and for area mm2. Please make the necessary corrections to the X-axes titles.

S9 Fig. 7: The "Raw data." can be removed from the figure title.

On a final note, one research prospect would be to qualify and quantify the resulting lithic flakes and flake scars morphometrics (2D and 3D) in order to establish if the use of bone versus hardwood produces any differences. This would be extremely helpful to circumvent the preservation issues of perishable materials and could in turn help lithicians establish whether the use of hardwood was common or not in Prehistoric times.

Reviewer #2: This paper reports experiments in using boomerangs to retouch the edges of flakes, as described in various Australian ethnographic accounts. The paper compares the stigmata of retouching on wood boomerangs to the stigmata produced on bone retouching tools. The study addresses an interesting topic and has the potential to be an important contribution to the literature, but, as currently written, it suffers from a lack of a clear research question. This, in turn, has led to a disorganised manuscript that attempts to do too much. I will describe the two main issues I have with the paper to consider in revision, followed by detailed comments I wrote as I read through the paper.

One issue is that the purpose of the paper—the research question—is not clearly articulated. In the conclusion we learn that the experiment was apparently to test whether boomerangs ‘can successfully function as retouchers’. But this is a non-question, as we already know from historical descriptions—reviewed in the senior author’s cited publications—that boomerangs were certainly used successfully by Aboriginal people to retouch stone tools. This is not in itself an interesting or useful new observation, and an experimental approach adds nothing to it. Another purpose was to explore whether bone and wood indentors sustain similar stigmata when used to retouch tools. This is a novel and worthy research question, but the authors conclude at the outset of the study (e.g., Lines 224-225) that the stigmata categories are the same for bone and wood, so it is a logical fallacy to nominate this as an experimental goal (Line 75) or outcome (e.g., Line 681 and elsewhere). We also learn late in the paper that the experimental protocol aimed to explore the spatial distribution of the stigmata on the boomerang in relation to the knapper’s gestures (e.g., Lines 655-657); again, an interesting and worth topic, but this is not developed in the methodology or background discussion. The lack of clarity at the outset of the paper results in a poor structure overall, which in turn leads to questionable decisions about what parts of the study are relevant (e.g., the bone retouching part of the study—this has already been done for European bone indentors; what is new here?) and which data need to be included in the main body of the text. These points are discussed in detail below, by line number. In revision, I suggest adapting some of the relatively more explicit statements of the intent of the study that appear in the conclusion, move them to the beginning, and be far more rigorous about what data is necessary to achieve the intent.

A second issue is that the paper’s statement of originality appears to be based on a lack of engagement with the literature. Specifically, ‘there is no archaeological, technological, or experimental evidence that wooden and osseous surfaces would have similar reactions to the percussive movement and the consequent impact with lithic edges associated with retouch activities.’ Experimental work in using wood retouching tools—for both pressure and percussion flaking—dates back nearly 100 years, if not earlier. For instance, there is an early video online of Coutier using wooden percussion flakers (perhaps written about in his papers in French), and Leakey was experimenting with wood also. An important early example is Crabtree’s 1970 manuscript (Crabtree, Don E. 1970 Flaking Stone with Wooden Implements. Science 169.3941: 146-153). There are many references to using wood indentors throughout the more recent experimental flintknapping literature, and some modern traditional flintknappers prefer them for some materials and techniques. And much of the published work explicitly compares bone (e.g., antler) and wood for the purpose of retouching stone tools. In Australia, Moore (2004) anticipated the authors’ research by describing boomerang retouching, and discussed the effects of soft-hammer (boomerang) retouching on the morphology of the resulting tula adze slugs. In revision, this literature—minimally, the Crabtree and Moore papers—should be cited, and the statement of originality revised accordingly. This paper is an important contribution, but it doesn’t need to be the ‘first’ to assert this contribution.

Line specific comments: (see attachment)

6. PLOS authors have the option to publish the peer review history of their article (what does this mean?). If published, this will include your full peer review and any attached files.

Reviewer #1: **Yes: **Luc Doyon

Reviewer #2: No

---

## [Author Response · Author response to Decision Letter 0]

13 Jun 2022

Ref.: Submission of revised manuscript PONE-D-22-05291

Dear Dr Spinapolice,

Thank you for considering our manuscript and for your email enclosing the reviewers’ comments.

We have reviewed the comments and have revised the manuscript accordingly. We found the comments very helpful for a significant improvement over our initial submission.

We incorporated most of the reviewers’ suggestions, except for the request of Reviewer #2 regarding the exclusion of the words for “boomerang” in original Aboriginal languages. We strongly believe that acknowledging the language names of a handmade object is a form of respect towards the cultural and creative significance of the object itself and, by extension, of the craftsman’s culture and creativity. We want to stress that the connection between an object, its maker and its maker’s tradition is a crucial concept in Australian Aboriginal culture.

Moreover, as it was a major concern for Reviewer #2, we restructured the Introduction section in order to state our research question more clearly. Parts of the Discussion section have been integrated accordingly. The section on experimental session 1 has been reduced to please the requests of both reviewers, and some key information in this section has been moved to SI.

We integrated the reference lists as requested by both reviewers, including more citations regarding previous studies on organic percussion retouch technology in Australia. In addition, we edited reference [2], which was under review at the time of our first submission and was only recently published. 

Finally, we addressed all comments regarding poor wording issues, and we revised the whole text to improve the flow of the paper. 

Appended below is our response to the reviewers’ recommendations. We hope that the revised version is now suitable for publication and look forward to hearing from you in due course.

Sincerely,

Eva F. Martellotta 

PONE-D-22-05291

Beyond the main function: An experimental study of the use of hardwood boomerangs in retouching activities

PLOS ONE

Dear Dr. Martellotta,

Thank you for submitting your manuscript to PLOS ONE. After careful consideration, we feel that it has merit but does not fully meet PLOS ONE’s publication criteria as it currently stands. Therefore, we invite you to submit a revised version of the manuscript that addresses the points raised during the review process.

We look forward to receiving your revised manuscript.

Kind regards,

Enza Elena Spinapolice, Ph.D

Academic Editor

PLOS ONE

Journal Requirements:

Reviewers' comments:

Reviewer's Responses to Questions

Comments to the Author

1. Is the manuscript technically sound, and do the data support the conclusions?

Reviewer #1: Yes

Reviewer #2: Partly

2. Has the statistical analysis been performed appropriately and rigorously?

Reviewer #1: Yes

Reviewer #2: No

3. Have the authors made all data underlying the findings in their manuscript fully available?

Reviewer #1: Yes

Reviewer #2: Yes

4. Is the manuscript presented in an intelligible fashion and written in standard English?

Reviewer #1: Yes

Reviewer #2: Yes

5. Review Comments to the Author

Reviewer #1: In their manuscript entitled "Beyond the main function: An experimental study of the use of hardwood boomerangs in retouching activities", Martellotta and colleagues investigate the potential of Aboriginal Australian boomerangs as implements to shape stone tools and compare the resulting use wear to that observed on bone retouchers which are well-documented in Palaeolithic and Neolithic contexts. To address this two-fold issue, they conceived and implemented a thorough experimental program, complemented with sound statistical analyses. I wish to emphasize here the diligent work produced by the authors and their apparent ethics in sharing ALL the data resulting from their experiments; the SI are very impressive. This will surely provide future researches on the subject with a wealth of data that can effectively be used in comparisons. Overall the manuscript is well-structured and is definitely pertinent for publication in PLOS ONE.

Below, I have only a few minor suggestions which would need to be addressed before publication. I am presenting it line by line.

Lines 113-114, 154-156 and 213-215: The equipment used for photography and video recording is repeated. Perhaps is can be avoided by adding a single general statement (at the beginning or the end of the Materials and Methods section).

Thank you for your suggestion. We replaced the references to the equipment with a general sentence at the beginning of the Materials and Methods section: “Pictures and videos of the experimental materials and sessions were recorded using the following equipment: a Canon PowerShot SX400 IS digital camera, a GoPro HERO7 White camera (v. 02.10) and a Canon EDS 800D digital SLR camera”.

Lines 127 and 145: The letter C is used twice for two different meanings: line 127 it refers to the circumference of the whole bone, line 145 it refers to the width of the arc of a bone fragments (according to Neruda, Lasznikova-Galetova, 2018). In addition, on line 145, H is not defined. They authors might want to re-word the lines 145 to 149 to make it more comprehensible.

Thank you. We realised that it is not necessary to use letters to refer to the measurements of the bones, and we have therefore deleted them from the text. Consequently, the letter C only stands for “radius of the arc”, following Neruda and Lázničková-Galetová (2018). As requested, we enriched the definition of C and H values and reworded the paragraph: “General measurements of the retouchers include: maximum length and width recorded according to the major axis of the bone blank; weight; external thickness; cortical bone thickness (S2 Fig b). The width of the arc (C) and the height of the midpoint of the arc (H) were measured in correspondence to the use area. Those measurements are necessary to calculate the radius of the arch to assess the role that the convexity of the surface plays in the retouch activity”. 

Line 188: the knapper T.R.M. is not an author, and realized who it was only after reading the Acknowledgments. Perhaps spelling their name when introduced the first time would be appropriate.

We corrected as suggested. Thank you.

Lines 258-259: This information was already presented in the Materials and Methods section. It s necessary to have it here?

We deleted the redundant information.

Lines 287-317 and 328-452: I have two small issues with these sections. First, in selecting only 4 of 14 retouchers for their description, they only present data from one knapper Y.L.P. The authors may argue that this is to allow comparison between bone and boomerangs, but it masks, if any, the (qualitative) variability between knappers. Second, the descriptions are somewhat redundant. The authors may wish to streamline these sections so the reader focuses more on the ensuing comparisons, which are absolutely necessary to answer the research questions stated in the introduction.

Thank you for your comment. Indeed, we chose those retouchers to offer a more consistent comparison with the boomerangs. However, we did not take into account the implications of masking the variability between knappers. In order to please the request of also R#2, who complained an excessive focus on bone retouchers in our experiment, we deleted these sections. Information regarding R6, R8, R12, R21, as well as previous S7 Figs 1-4, have been moved to S6_File. We reworded the section ‘Retouching session 2 - Hardwood boomerangs’ as requested.

Line 288: "found both areas sufficient", please be more explicit.

We deleted this section of the text. Please refer to the comment above.

Line 310: "was exploited in one portion", please clarify, in a single session?.

We deleted this section of the text. Please refer to the comment above.

Line 320: I think the first sentence can be removed as it is seldom pertinent here to say the boomerangs were experimentally used for the same function as bone retouchers...

Thank you, we corrected as suggested.

Lines 466-476: ensure the proper math sign and unit conventions are used throughout this paragraph.

Thank you for pointing it out. We modified this section by using the symbol “σ” for the standard deviation. We also used a single space before and after the equality symbol. Finally, we corrected the superscript ‘2’ in mm2 when necessary. 

Lines 526-560: I have a difficulty to understand how the percentage were calculated. It would appear the authors report here the sum of one mark type divided by the total number of marks of all types. This is slightly misleading as one blow can produce multiple marks (e.g., R10_1 with 4 blows that produced 58 marks) and another can produce none (e.g., R10_2 with 46 blows and only 10 marks or R11_1 with 12 blows and only 5 marks).

Thank you for your comment. We indeed use this methodology to calculate the percentages. We believe it is an efficient way to give the reader an idea of how many linear and punctiform impressions occur during the retouch activity without necessarily going through the table. In order to clarify, we specified in the paragraph that those percentages refer to the total amount of identified stigmata. Moreover, we added the following statement: “Previous experimental studies on the use of bone retouchers proved that the association between the number of blows and stigmata is highly variable. Such proportion never reflects perfect equivalency (i.e., 1 blow = 1 stigma), making the retouch activity susceptible to several variables”.

Lines 555-556: This statement should be statistically tested rather than assumed, especially since the authors have all the data available.

Thank you. We added a scatter plot showing the negative correlation between linear marks and notches (S9 Fig 5). Moreover, we replaced the misleading phase “to assume” with the more accurate “It is possible to test the existence of a negative proportional correlation between linear impressions and notches”.

Line 572: "are often interact with other", poor wording.

We corrected as follows: “Scraping marks are shallow and long, and they often interact with other stigmata”.

Lines 576-577 and 737-738: Similar "peeling" can occur when weathered bone is used in knapping activities (see Mallye et al., 2012, Doyon et al., 2018, 2019).

We are grateful to the Reviewer for suggesting these references. We have now included them within the text.

Lines 580-583: The authors may want to delete these was the information is presented in the section "Embedded micro-flakes".

As suggested, we moved the sentence in the “Embedded micro-flakes” section.

Lines 591-594: It is not clear why this was not presented in the Materials and Methods. It is not clear either where the maximum of 256 measurements recorded on bone retouchers comes from. And, as a consequence, it is difficult to interpret the figures presented in S9 Figs. 5-6.

A significant difference in frequency exists between stigmata on bone retouchers (2115 linear, 256 punctiform) and boomerangs (6361 linear, 1651 punctiform), and it created some problems in the graphic representation of linear and punctiform impressions’ metric data. Those problems are shown in ex S9 Fig 5 (now S9 Fig 6), and we wanted to give back a more balanced representation of stigmata’s metric data. To solve this problem, we manufactured a new database containing the length of linear impressions, and surface and perimeter of punctiform impressions. Because the discrepancy in frequency was due to the greater number of observations in the boomerangs’ database, we decided to shorten this database and make it of the same length of the bone retouchers’ database, which contained a minimum of 256 observations. The shortening process was carried out by selecting 256 random observations from the original database of the measurements of stigmata on boomerangs. As a result, we now have two databases of the same length, which can be compared quantitatively (represented in the new S9 Fig 7). As requested by R#1, we moved the sentence regarding the applied methodology to the Materials and Methods section. Moreover, as requested by R#2, we added some scatter plots (S9 Figs 8-10) to give a sense of the (low) statistical significance of this comparison and emphasise why we initially chose only a quantitative representation instead. 

Line 627: please cite some references for the archaeological and ethnographic research.

We integrated this section with more references and discussions: “The multipurpose nature of boomerangs is a well-established concept in Australian Aboriginal Traditional knowledge, although it has received little direct investigation by archaeologists and ethnographic researchers [e.g. 2,6–9,21]. In our recent work [2], we performed a systematic quantitative review of the literature available on the subject of “boomerangs”. Our analysis showed how most previous boomerang-focused publications mainly consider the aerodynamic properties linked to the boomerang’s infamous returning abilities. Technological and functional aspects of non-returning boomerangs -- including their use as retouchers -- can be found in the form of incidental citations within broader descriptions of the Indigenous Australian lifestyle and daily activities (e.g., woodworking) [2]. Consequently, the remarkable variability of non-returning boomerangs is frequently overlooked.

During the second half of the 20th century, ethnographic reports on boomerangs are present in the literature, but they seem to stagnate in approaches and references belonging to the beginning of the century. Although their contribution to the current knowledge on boomerangs has to be recognised, it could also be argued that they are nowadays less suitable to contribute to a multidisciplinary and more scientific approach to ancient technology [2]. Suffice it to know that Davidson’s work of 1936 [45] currently remains the main comprehensive study on boomerangs. Davidson himself concluded his contribution (p. 90) by wishing for an approach to boomerangs that was more technology-inclined than typological [2,45].

Regarding recent contributions boomerangs have been rarely studied: this is a relevant issue, considering how much theories and techniques for the study of ancient technologies evolved in the last 30 years. Nevertheless, in this timeframe boomerangs have only been the object of sporadic classifications [7,25,46]; in other cases, they played a marginal role in valid experimental programs [21,46]; finally, they were summoned when rare archaeological remains of boomerangs were discovered [15,16]. This approach has overshadowed the role that these tools have played, and keep playing, in Australian Indigenous societies”.

Lines 632, 650, 655, 687, 755: The authors refer to discussion or figures from other publications. Reproduction of these figures (if rights allow it) and synthesis of the main points relavent to the discussion should be included in the text rather than simply referring to other works.

We included this picture in the SI (new S7 Fig) and referred to it accordingly during this all section.

Lines 692-693: "repetitive impacts [...] repeatedly fracture" , the wording is a little off.

We deleted the word “repetitive”.

Line 707: "A pattern emerges when it these aspects are..." , wording issue.

We corrected as follows: “a pattern emerges when these aspects”.

Lines 709-710: it is not clear if the authors refer to assemblages containing few lithic tool types or few lithic retouched tool specimens.

We rephrased to clarify that we refer to the presence of few lithic retouched specimens: “However, a pattern emerges when these aspects are compared, suggesting that lightly used bone retouchers are associated with assemblages containing a low number of retouched lithic tools”.

Lines 716-730: Perhaps it would be pertinent to include the references from Jéquier et al 2018 and Doyon et al 2019 on this subject and see how their results compared to those presented in this study.

We are grateful for the suggestions. We enriched the references accordingly.

Lines 797-798: "demonstrate that osseous and wooden percussor tools have physical and mechanical properties in common" , although this is not completely wrong, the physical and mechanical properties were not tested per se. What the experiment shows is that hardwood tools can be used as retouchers just as bone, and that the resulting use wear is strongly similar.

We agree with the Reviewer. It is a subtle, however important, specification. We modified this section as follows: “Moreover, this work represents one of the first experimental studies to demonstrate that osseous and wooden percussor respond similarly to percussion stimuli in the context of retouch activities. Several traceological similarities among the stigmata strongly support this conclusion”.

Lines 816-952: Ensure the reference section matches the guidelines for authors.

We revised the refence list. Reference [2], still in review at the time of our first submission, is now published. We cited it as follows:

Martellotta EF, Brumm A, Langley MC. Tales of Multifunctionality: a Systematic Quantitative Literature Review of Boomerangs Used as Retouchers in Australian Aboriginal Cultures. J Archaeol Method Theory. 2022; 1–25. doi:10.1007/S10816-022-09561-X

Tab. 1: some formatting issues can be resolved easily, i.e., titles, numbers on two lines after the points, etc.). It was not clear, however, why the use areas were separated into distinct lines, especially since the information is expected to be the same. Rather than duplicate the lines, I would remove the information on use area name, AND add a column on the number of use area per retoucher. You might want to transfer the column (n. of use areas) from Tab. 3 to Tabs. 1 and 2 where it is more pertinent.

That is an excellent point. We corrected Tables 1 and 2 as suggested and resolved the formatting issues in Table 1.

Tab. 2: you might want to remove the "_1" from the tool ID column.

We corrected as suggested.

Tab. 4: Formatting issues. Line order should be R1, R2, R3, R4, etc.

We solved the formatting issues and redistributed the line order.

S7 Figs. 1-4 (panel lithic tool): it would be interesting to highlight (perhaps only with a dash line) the location of the retouch along the edges.

We deleted this text section and the associated figures (please see pertinent comment above). However, we applied the Reviewer’s suggestion to Figures 1-5. Thank you.

S9 Fig. 1: The Y-axes are not the same for boomerangs and bone retouchers. Perhaps, the two tool types can be compared side-by-side for each metric.

Thank you; we improved the plot as suggested (new S9 Fig 7).

S9 Figs. 3, 4: There are some discrepancies between the data shown in the figures and those presented in Tab. 3. In Tab. 3, the maximum linear impressions are 559 and 432 for boomerangs and retouchers. The maximum number of punctiform impressions is 209 and 69. The title of S9 Fig. 4 should state "scatterplot" rather than "statistical distribution". The % of the confidence of intervals should be indicated.

We are grateful to the Reviewer for their keen eye. Indeed, Table 3 in the manuscript has been cut in half, and some rows disappeared. Probably some formatting issue with MS Word… We now integrated the table with the missing information; it is now consistent with the data represented in S9 Fig 3. As suggested, we changed the title of S9 Fig 4 from “statistical distribution” to “scatter plot”. Finally, we indicated in the caption that the percentage of the confidence of intervals is 95%.

S9 Fig. 5: the unit for perimeter should be mm and for area mm2. Please make the necessary corrections to the X-axes titles.

We corrected as suggested. Thank you.

S9 Fig. 7: The "Raw data." can be removed from the figure title.

We corrected as suggested

On a final note, one research prospect would be to qualify and quantify the resulting lithic flakes and flake scars morphometrics (2D and 3D) in order to establish if the use of bone versus hardwood produces any differences. This would be extremely helpful to circumvent the preservation issues of perishable materials and could in turn help lithicians establish whether the use of hardwood was common or not in Prehistoric times.

We are grateful to the Reviewer for this helpful suggestion. Indeed, it is a fascinating, although scarcely investigated topic within the retouching field, and we decided to improve the “future research” section as follows: “Finally, as recent studies [59,60] proposed methods to distinguish various types of soft hammers through the study of lithic detachments and scars, it could be rather intriguing to include wooden retouchers in this discussion”. 

 

Reviewer #2: This paper reports experiments in using boomerangs to retouch the edges of flakes, as described in various Australian ethnographic accounts. The paper compares the stigmata of retouching on wood boomerangs to the stigmata produced on bone retouching tools. The study addresses an interesting topic and has the potential to be an important contribution to the literature, but, as currently written, it suffers from a lack of a clear research question. This, in turn, has led to a disorganised manuscript that attempts to do too much. I will describe the two main issues I have with the paper to consider in revision, followed by detailed comments I wrote as I read through the paper.

One issue is that the purpose of the paper—the research question—is not clearly articulated. In the conclusion we learn that the experiment was apparently to test whether boomerangs ‘can successfully function as retouchers’. But this is a non-question, as we already know from historical descriptions—reviewed in the senior author’s cited publications—that boomerangs were certainly used successfully by Aboriginal people to retouch stone tools. This is not in itself an interesting or useful new observation, and an experimental approach adds nothing to it. Another purpose was to explore whether bone and wood indentors sustain similar stigmata when used to retouch tools. This is a novel and worthy research question, but the authors conclude at the outset of the study (e.g., Lines 224-225) that the stigmata categories are the same for bone and wood, so it is a logical fallacy to nominate this as an experimental goal (Line 75) or outcome (e.g., Line 681 and elsewhere). We also learn late in the paper that the experimental protocol aimed to explore the spatial distribution of the stigmata on the boomerang in relation to the knapper’s gestures (e.g., Lines 655-657); again, an interesting and worth topic, but this is not developed in the methodology or background discussion. The lack of clarity at the outset of the paper results in a poor structure overall, which in turn leads to questionable decisions about what parts of the study are relevant (e.g., the bone retouching part of the study—this has already been done for European bone indentors; what is new here?) and which data need to be included in the main body of the text. These points are discussed in detail below, by line number. In revision, I suggest adapting some of the relatively more explicit statements of the intent of the study that appear in the conclusion, move them to the beginning, and be far more rigorous about what data is necessary to achieve the intent.

We are grateful to the Reviewer for their helpful insights. We believe most of the Reviewer’s confusion is due to a poor statement, on our side, of our previous works on this topic. As we stated in the Introduction, the present experiment is the continuation of ongoing research composed of (1) literature evidence of boomerangs used in percussion retouch, (2) identification, through a systematic literature review and a lexical analysis of the literature, of similarities between the use of boomerangs (and other wooden tools) and bone retouchers, (3) identification, through traceological analysis, of retouch-induced stigmata on museum curated boomerangs, (4) the assumption that wood shares some properties with bone and they could therefore function in similar ways and similarly react to percussion. (1) and (2) are addressed in our previous work [2], whereas (3) is addressed in [1]. Our previous studies pointed to the same hypothesis: a technological parallel with bone retouchers could improve our understanding of the boomerangs used in retouching activities. The present work consists of experimental validation of this hypothesis.

Please refer to the comment below for our response to the Reviewer’s suggestion on previous literature. Thank you.

In order to accommodate the reviewer’s requests, we rephrased some crucial parts of the Introduction to state our research question more clearly. We added more references regarding the previous ‘experimental’ use of boomerangs, and we openly explained why we feel a new protocol is needed. Similarly, we state how our new protocol relates to literature evidence on the use of boomerangs as retouching tools. Moreover, we clarified the misunderstanding regarding the assumption of similarities between use wear on boomerangs and stigmata on bone retouchers at the beginning of our study. More than an assumption, it is a hypothesis based on traceological preliminary studies. Finally, as requested, we added more information regarding the knapper’s gesture, including more details in relation to the spatial distribution of the use wear on the boomerang’s surface. All the above is discussed in the replies to the Reviewer’s specific comments below. Thank you.

A second issue is that the paper’s statement of originality appears to be based on a lack of engagement with the literature. Specifically, ‘there is no archaeological, technological, or experimental evidence that wooden and osseous surfaces would have similar reactions to the percussive movement and the consequent impact with lithic edges associated with retouch activities.’ Experimental work in using wood retouching tools—for both pressure and percussion flaking—dates back nearly 100 years, if not earlier. For instance, there is an early video online of Coutier using wooden percussion flakers (perhaps written about in his papers in French), and Leakey was experimenting with wood also. An important early example is Crabtree’s 1970 manuscript (Crabtree, Don E. 1970 Flaking Stone with Wooden Implements. Science 169.3941: 146-153). There are many references to using wood indentors throughout the more recent experimental flintknapping literature, and some modern traditional flintknappers prefer them for some materials and techniques. And much of the published work explicitly compares bone (e.g., antler) and wood for the purpose of retouching stone tools. In Australia, Moore (2004) anticipated the authors’ research by describing boomerang retouching, and discussed the effects of soft-hammer (boomerang) retouching on the morphology of the resulting tula adze slugs. In revision, this literature—minimally, the Crabtree and Moore papers—should be cited, and the statement of originality revised accordingly. This paper is an important contribution, but it doesn’t need to be the ‘first’ to assert this contribution.

Thank you for your comments. We believe the suggestions made by the Reviewer are valid. However not exactly suitable for our study. Crabtree’s work, for instance, addresses the topic of pressure retouch using wooden tools, whereas we stated several times in our manuscript that our work aims at percussion retouch only. It is important to keep it in mind when we state that the technology behind only some retouched industries -- produced by a percussion retouch -- is less investigated by Australian research than pressure-retouched tools.

On a more general note, we believe the Reviewer’s suggestions indirectly improved our manuscript because they gave us the opportunity to make the Introduction more pointed. However, we believe that the suggested previous works are not suitable for a direct comparison with our work. Although they all are surely important contributions to the experimental study of wooden percussors, it could be argued that their methodological approach is, at the moment, relatively obsolete compared to what experimental archaeology is nowadays. Coutier, Crabtree, Leakey, and others’ approaches have been revolutionary. However, they happened in a moment when experimental archaeology was at an embryonal stage, often edging into imitative archaeology, heavily influenced by anthropology, and almost completely disconnected from scientific-inclined disciplines, such as traceology. It is a crucial distinction to keep in mind when summoning a methodological comparison with the cited contributions. We would like to specify that we do not include Moore (2004) in the same category. His work indeed represents a different, more up-to-date methodological approach to the subject. Although his results are not directly compared to ours -- only lithic analysis was undertaken in Moore (2004), and no traceological analysis of the boomerangs appears in the results -- we believe it should definitely be included as a reference in our study. Please see specific comments below for a more detailed response. Thank you.

To accommodate the Reviewer’s requests, we restructured the Introduction in order to (1) clarify our research question, and (2) acknowledge previous studies on the matter of retouching with wooden tools and relate previous contributions to ours. More detailed information on our integrations within the text can be found in our responses to the Reviewer’s specific comments. Thank you. 

Line specific comments: (see attachment)

Line 48. Awkward phrasing. E.g., ‘retouching techniques is lacking in the current research interest’. The paper has awkward phrases periodically throughout, and could benefit from a careful language edit. Some of my confusion or misunderstanding may result from issues around expression.

We apologise for some poorly written passages of the paper. We rephrased here (please see comment below) and wherever else was necessary. Thank you.

Lines 45-50. ‘Retouched lithic tools’ needs defined. The claim about a lack of systematic investigation of ‘tools shaped by means of percussion retouching techniques’ is poorly justified; there are lots of systematic investigations of just this. Indeed, it seems to me that the vast majority of lithic studies in Australia are focused entirely or in part on retouching, either in shaping tool types or gauging reduction intensity by retouching throughout a tool’s uselife. The authors have not demonstrated a gap in the literature, as written. It seems to me that part of the issue here is that the study may be adopting a wider European definition of retouching, rather than a more narrow Australian/North American definition.

We are grateful to the Reviewer for giving us the possibility to improve this crucial section of our manuscript. Keeping in mind that we refer to the study of percussion retouch only, it could be easily argued that the interest in the technology of retouching (i.e., types of retouching tools, materials for retouching tools, traceology of retouching tools, techniques, geographical variability, and more) is much more prominent in the study of pressure retouch rather than percussion retouch. It does not undermine that Australian research history shines in the study of lithic tools and retouching. However, as we said in the Introduction, pressure and percussion retouching technology have a different weight in the research interest -- here intended as, for instance, number of recent publications, quality of publications, and multidisciplinary approaches applied to publications. In order to clarify this concept, we restructured and integrated the whole Introduction section. Please see also the comment below. Thank you.

Line 68: no experimental evidence that wooden and bone would react similarly in percussion flaking. Crabtree 1970 and maybe elsewhere in his work. Coutier, Leakey, Clark, possibly Pelegrin, Bradley, Newcomer. Kamminga?

In order to clarify, we added the following in the Introduction: “In Australian archaeology, few studies have investigated potential similarities, technologically speaking, between osseous and wooden objects [12, and references therein]. This lack of studies is mainly owing to the rarity of wooden items recovered in archaeological contexts because of harsh environmental conditions [14,15, and references therein]. On the other hand, most of the knowledge of Australian wooden tools comes from ethnographic sources. Among these, evidence of wooden implements -- especially boomerangs -- employed in percussion retouching of lithic tools is rare but present [2,6,8,9,17-22,23,24]. However, none of those contributions engages in an experimental protocol aimed at investigating the technology behind the use of boomerangs in retouching activities. An exception is [21]: one of the few recent contributions addressing, experimentally, the topic of boomerangs used to shape stone tools; this work, however, mainly focused on the retouched stone tools (Tula adze) rather than on the boomerangs per se. As a result, the technological implications of the use of wooden boomerangs in retouching activities constitute a gap in the current research. If and how those implications could be compared with osseous tools used as retouchers is an even less investigated topic”.

Lines 74, 76: How is ‘function well’ as retouchers to be measured in this study? Also, how is ‘to easily reproduce’ to be measured? Neither are, in fact, measured or evaluated by this study that I could see.

We agree with the Reviewer on our poor wording in this section. We replaced the vague concept of “function well” with a more precise phrasing: “The questions addressed with this experiment are (1) which variables are involved in the use of boomerangs to retouch lithic tools? (2) how does the resulting use wear compare to that observed on bone retouchers? We expect to easily reproduce, using boomerangs, intensively retouched lithic flakes reflecting morphologies found in the archaeological records”. Thank you.

Line 83: The reference to Palaeolithic Europe is a bit of a clanger which points to a Eurocentric bias. Why not Africa? South Asia? America? This also brings up another niggling doubt I have about the usefulness of the paper beyond wooden percussion-flaking tools, which are, to my knowledge, absent from the archaeological record and only known from the Australian ethnographic record world-wide. How can detailed knowledge of the stigmata on wood be bootstrapped to a larger issue (one obvious way is correlating with the ‘stimata’ on stone flakes and retouched edges, as already done by Moore 2004 for boomerang-retouched adzes)?

Thank you for your comment. As we explain in the Introduction, this experiment is part of ongoing research which produced the hypothesis of a technological parallel with bone retouchers. We broadly expressed the reasons behind it in [1,2]. We believe restating here the discussions of our previous works might be redundant. However, we reworded this section of the Introduction with more details: “In our previous works, preliminary investigating this issue [1,2], we proposed the use of hardwood boomerangs for retouching purposes among Indigenous Australian communities until at least European incursions. Our hypothesis -- and the foundation of the present experiment -- was based on four central notions: (1) boomerangs hold a deep multipurpose value in Australian Aboriginal societies [e.g., 2, 6–9]; (2) literature evidence proved that the use of boomerangs (and other wooden tools) in retouching and resharpening activities could be technologically associated to the use of Palaeolithic bone retouchers [2]; (3) a traceological analysis of museum-curated boomerangs identified the presence of retouch-induced stigmata comparable with the ones identified on bone retouchers [1]; (4) bone and wood, as materials for tool-making, share some mechanical and physical properties, and it is possible to hypothesise they would have similar reactions to the percussion movement in retouching activities”.

Moreover, we believe that the absence of wooden tools in the archaeological record should not discourage future research from investigating their technology. It seems reductive to axiomatically exclude the use of wooden tools in Prehistory and different parts of the world only because most of the material record comes from the ethnographic era in Australia. Wooden tools in archaeological contexts have been recovered in different sites -- Schöningen (300ky BP), Poggetti Vecchi (171ky BP), Aranbaltza (90 ky BP) -- and although they have none of them contains traces of percussion activities, it does not mean that those activities were not performed. Several researchers are, in fact, convinced that organic tools in Prehistory could have had a much more relevant role than lithic tools and that our perception of ancient technology is influenced by the (poorly) preserved material record. From this point of view, we believe our work could contribute to an international debate regarding the role of wooden tools in ancient technology.

Line 87. Methodology is inadequate. How do these articulate with the research question? What exactly is the research question? Why is the manufacturing process for bone retouchers (e.g., Line 116 onwards) important to the research, but manufacture of the boomerangs apparently is not? Also, methodologically-relevant material occurs very late in the text, rather than up front where it belongs. There are vague indications of evaluating various things, but with little information on how (or if) this was done (e.g., see above, Lines 74, 76).

Thank you for your suggestions. Please refer to the comments above for the adjustments we did on the Introduction to clarify the research question and the bone retouchers' role in our protocol. Moreover, we enriched this section of the methodology as follows: “The experimental program involved four stages: manufacturing lithic flakes, manufacturing bone retouchers and hardwood boomerangs, retouching lithic flakes with bone retouchers, and retouching lithic flakes with boomerangs. Subsequent to the retouching sessions, the resulting use wear was analysed and compared by creating a database and applying basic statistic tests using the software R [26]. In the measurements of some specific use wear, the ‘sample’ function in R software was used to generate a random subset of data, more suitable for statistic correlations. Pictures and videos of the experimental materials and sessions were recorded using the following equipment: a Canon PowerShot SX400 IS digital camera, a GoPro HERO7 White camera (v. 02.10) and a Canon EDS 800D digital SLR camera”.

As suggested by the Reviewer, we moved within the methodology section some sentences which were initially located in the Results/Discussion sections. Finally, we added a new paragraph on manufacturing the boomerang replicas used in our experiment (please see specific comment below).

Line 94. The company name is ‘Flintknapping Supplies’

Thank you. We corrected accordingly.

Line 102. The authors clam that, in relation to material types, ‘variations in stigmata’s features are statistically and qualitatively not relevant’ (Lines 102-103). I was surprised by this comment, as the authors of the cited reference conclude the precise opposite: ‘the marks produced on retouchers differs depending on whether they were used to strike flint or quartzite’ (Mallye et al. 2012, quote in the abstract). Also, ‘In effect, the scores with smooth surfaces are mainly associated with objects used to retouch flint flakes while those with rough surfaces are associated with objects used to retouch quartzite flakes (chi-square calc . 26.98; P < 0.001)’ (Mallye et al. 2012:1136). And ‘we found that the combination of the morphology of the traces and the types of use areas allows us to deduct the type of raw material retouched’ (Mallye et al. 2012:1137). This is a key finding of the Mallye et al. study. Also, material types aside, the authors’ study is comparing the marks on wood to bone, so assuming that the experimental results can be transferred from bone to wood at the outset is not very useful and undermines the paper’s logic.

We apologise if our wording was misleading in this section. We meant that, although the use of different lithic raw materials could cause some variations within the retouch-induced stigmata, those variations are qualitatively irrelevant in the context of our study. That is, retouch-induced stigmata created by a flint edge are distinguishable from those created by a quartzite edge, but they are more similar to each other than to a cutmark. Moreover, we would like to specify that our traceological analysis is based on several works -- Mallye et al., Mozota Holgueras, Tartar, -- and our previous traceological study of museum-curated boomerangs [1].

In order to clarify, we added the following: “. Although previous experimental studies of bone retouchers revealed some differences in the use wear’s features when retouching either coarse or fine lithic materials [27], we believe it is a statistically and qualitatively not relevant issue in the context of our study”.

Moreover, we reworded and integrated the section “Materials and Methods -- Retouch-induced stigmata analysis” as follows: “The use wear resulting from the impact between the lithic edge and the osseous/wooden retouching tool’s surface is defined as “retouch-induced stigmata”; the stigmata group in small portions of the retoucher’s surface, defined as “use-areas” [27,33]. In our experimental sample, stigmata were counted and grouped into four morphological categories based on previous works on bone retouchers [27,33,40]: (1) linear impressions are elongated, deep marks; (2) punctiform impressions are triangular or ovoidal depressions; (3) striations are short, shallow, and often parallel marks; (4) notches are deep and wide detachments of a small portion of the organic surface during an intense retouch activity. Finally, scraping marks can be present: they appear as long, shallow, linear marks covering a significant portion of the surface. The four morphologies of stigmata usually appear together, often overlapping, in the same use area. Their interaction generates four categories of “intensity of retouch”: isolated, dispersed, concentrated, and superposed [27,33]. In our previous traceological study on boomerangs [1], we proposed that those use wear, with similar features, also occur on boomerangs used in retouching activities. To verify this traceological evidence, we compared retouch-induced use wear on boomerangs with stigmata on bone retouchers in our experimental sample”.

Line 151. What is meant by a ‘low dash and a crescent number’? Clarify.

We rephrased as follows: “Multiple use areas on the same tool are indicated by the underscore symbol and ascending numbers following the Retoucher ID (example: R10_1)”. Thank you.

Line 160-161. The boomerangs were made ‘using traditional methods’. This needs to be expanded and clarified. I doubt very much that they were made using stone tools (in the archaeological terms relevant to this study, this seems to be the criteria for ‘traditional’ manufacture). To my knowledge, manufacture of boomerangs with stone tools no longer occurs in Australia. If that is correct, then the experiment’s boomerangs were made with iron or even power tools (which in my experience is universal for modern boomerang manufacture in eastern Australia)—and the marks on the boomerangs in the photographs certainly look like power tool marks, combined with wood rasp marks. This is an exceptionally important issue, as I have found that boomerang manufacture with stone tools—particularly with tula adzes used in the traditional adzing gesture—can create stigmata just like those documented in this study, including chert microflakes embedded in the adze chop marks. In some ways, boomerangs made by modern tools might be an experimental advantage, as it eliminates this source of variation by creating a smoother, unambiguous surface. Conversely, it makes applying the observations in this paper to archaeological or ethnographic tools more difficult, because the issue of sorting stone tool manufacturing marks from retouching usewear stigmata is not addressed.

Thank you for your comment. We believed that ‘traditional methods’ was self-explanatory because it is well-known that Aboriginal people stopped manufacturing boomerangs with stone tools since the first European incursions, through a process of integrating their technology with the new materials available. We believe, however, that giving more information regarding the manufacturing of our experimental replicas could improve our manuscript: “Before the first European incursions, Aboriginal peoples across Australia applied various methods and techniques to manufacture boomerangs, most commonly using stone tools. Boomerangs were made from different types of wood, including tree trunks, elbow bend branches and tree foot roots. At the first stage of manufacturing, a hafted stone axe was used to retrieve the desired section of the tree. During the following stages, stone or shell scraping tools were used to create a pre-form. The next step could either involve the hardening of the boomerang over hot coals or its soaking and twisting in water -- depending on the sought-after aerodynamic features. Finally, the surface was sanded smooth, often with sandpaper fig leaves. After the European incursions, Australian Aboriginal peoples had access to new materials (e.g., steel tools) which proved to be popular as they resulted in a faster, easier and more efficient way to shape wooden tools [some recent contributions to the manufacturing process could be found in 2,7,14,16,28, and references therein]. Four hardwood boomerangs were used in the present experiment, identified with the letter B and a sequential number, and manufactured by two expert Indigenous Australian artisans. These boomerangs were made as usable weapons using modern steel tools, including tomahawk, rasp files, sandpaper and electric 4-inch grinders. These modern techniques mimic, but greatly enhance, the speed and efficiency of the traditional methods used by the two artisans’ ancestors. As per experimental protocol, a screening of the manufacturing marks on the boomerangs’ surface has been carried out before the retouching sessions to ensure a proper distinction between manufacturing marks and use wear”.

Regarding the second issue raised by the Reviewer, regarding the manufacturing marks to be confused with use wear -- It is difficult to make the sort of comparison suggested by the Reviewer when there are very few recent references on traceological studies on boomerangs [25]. However, it is a rule in every experimental protocol to perform a screening of the surfaces involved in the experiment before the experiment takes place. This operation was performed in our protocol as well, and it is now written in the manuscript. Thank you. 

Linte 166. Mr Dennis is descended from a coastal SE NSW group (Yuin) and a western NSW group (Wailwaan), but the words suggested for boomerang range from different western NSW groups (Ualaroi, or Gamilaraay/Yuwaalaraay, and Wiradjuri). The ‘of this area’ in line 167 might be changed to ‘of Mr Dennis’s people’ or similar, as it presently implies that the language names are from SE NSW. The language names for boomerang are not relevant to the study in any case, so this section could be deleted entirely.

We strongly disagree with the Reviewer’s statement regarding the relevance of the Indigenous words for “boomerang”, and we believe it is very unfair towards the traditional knowledge shared with us by Mr Dennis and Mr Craft through their handcrafts. We strongly believe that acknowledging the language names of a handmade object is a form of respect towards the cultural and creative significance of the object itself, and by extension, of the craftsman’s culture and creativity. In fact, we believe that if more archaeologists had this type of sensitivity, collaborations with Aboriginal Australians would happen much easier, and the gap between “Western archaeology” and Traditional Knowledge could be narrowed. 

In the specific case raised by the Reviewer, when we commissioned the two boomerangs from Mr Dennis we received a certificate of authenticity, composed and signed by Mr Dennis himself, in which he states the words for “boomerangs” in the languages that he believed were relevant. We stand by our decision to respect Mr Dennis’ acknowledging of the Gamilaraay/Yuwaalaraay, Wiradjuri and Wailwaan languages because, as Mr Dennis himself stated in the document mentioned above, “[this] authentic and original […] craft piece [i.e., the boomerang] is representative of my authentic cultural and spiritual beliefs”. Thank you.

Line 181. use ‘varies’ rather than ‘goes’

We corrected as suggested. Thank you.

Line 186. Description of the retouching gesture(s) is necessary here. For instance, Mallye et al. (2012:1133) describe a ‘circular percussion trajectory’ in their experiments with bone retouchers. (I return to this several times below). Is this the same gesture used in this study?

Thank you for your comment. We integrated this section as follows: “The retouch activity was carried out by applying a percussion movement following an elliptical trajectory with a tangential point of impact”.

Line 191. The knappers must have had some effect they were attempting to achieve in retouching. At what point was their ‘arbitrary satisfaction’ met? Was it when the edge was completely retouched, for instance? Expand this a bit. (I see that this is covered in the concluding sections of the paper. Move it up front, to the methodology).

As suggested, we moved the information to the Methods section. Thank you.

Table 3. I began to become confused here, as Table 3 is the result of the experiments but it appears in the methodology section. This is premature, as not all of the methods and definitions reported in the table have been presented. Move Table 3 to the results section. I think tables 1 and 2 can stay where they are, as they are describing the tools used in the experiments, not the results of the experiments per se.

As suggested, we moved Table 3 in the Results section. Thank you.

Line 198. Some detail is necessary here on the types of gestures used by YLP to retouch the chert flakes using the boomerang. I see in the Fig 1 caption referent to a ‘tangential movement of retouch’, but I’m not sure what this means. The figure suggests that the movement was what I would describe as ‘flat’ with the face of the boomerang slapped onto the edge to be retouched, with minimal follow-through. This is different from the gesture photographed and described in Roth, and Roth’s photos in Moore (2004:Fig 2; not cited in the ms, and the only photograph I am aware of showing the use of a boomerang in retouching a stone tool—it should be cited), where the edge or face of the boomerang is swung and an oblique orientation to the tool’s edge, thus allowing follow-through. I suggest a new figure, or an addition to Fig 1, that schematically represents the gestures used by YLP in the experiments, and the terms used to describe those gestures. (The video shows YLP using a gesture somewhere between ‘slapping’ implied by Fig 1 and the very oblique method implied by the historical photograph in Moore 2004:Fig 1. Indeed, the somewhat puzzling technique in Camooweal of hafting the tula at right angles to the handle, specifically for boomerang retouching, may in fact have been done to allow this rather oblique gesture, thus allowing more follow-through than allowed by YLP’s gesture).

We are grateful to the Reviewer for their helpful insights. We would like to specify that we are familiar with the picture mentioned by the Reviewer -- that is, an unnamed Aboriginal man retouching the stone Tula adze using a boomerang (original photo from Roth, negative V2165, Division of Anthropology, Australian Museum), reproduced in Moore (2004, Fig. 2) and McCarthy (1961, Fig. 1). We largely cited this picture in our previous work regarding the literature evidence of boomerangs used as retouching tools [2]. However, when we sought permission to reproduce this picture from the Traditional Owners, they (understandably) denied it because their Country was not directly involved in the study. We strongly believe we should not reproduce a picture of a deceased man without permission, and for this reason, the photo does not appear in our work. To accommodate the Reviewer’s request, we added more references in the Introduction and Discussion about literature evidence of boomerangs used as retouchers [2, 6, 23,24,8,9,17-22].

Regarding the second issue raised by the Reviewer -- the movement applied during the retouch activity -- we implemented and uniformed the definition of the retouching movement throughout the manuscript. That is, a percussion movement following an elliptical trajectory with a tangential point of contact. We also specified that the movement applied to the use of boomerangs is the same applied to the use of bone retouchers. A small difference has been highlighted: a smaller diameter of the ellipse outlines the movement itself, which we believe is due to the bigger size of the boomerangs than the bone retouchers. Thank you.

Lines 224-225. The authors state that the categories of usewear documented are ‘valid for both bone retouchers and boomerangs used as retouchers’, but isn’t this what the experiments were meant to assess? There are some problems with the epistemology here. If the marks are known to be the same a priori, then what was the point of the experiments? This is a fundamental issue with the paper, here and elsewhere.

We modified this section in order to clarify. Please refer to the similar comment above. Thank you.

Lines 225-229. I suggest a new composite figure here as a guide, with an example of each mark labelled. The definitions are vague, using descriptors such as ‘deep’ and ‘wide’, so some photographs would help here.

We rephrased this section and inserted a more explicit reference to previous traceological studies on bone retouchers and boomerangs, which include detailed figures describing the stigmata. Please see also similar comment above. Thank you.

Lines 230-238. More clarity is needed here. How was the 3D modelling done? Was it generated by the microscope by image stacking, or what? What was the instrument used? What scales were used? What is the margin of error for measurements? What are the ‘equal distances’ referred to?

Thank you for your comment. We would like to specify that we have been working with 3D scans and not 3D modelling, the latter being a technique not suitable for our study. The procedure we applied to obtain the 3D scans of single use wears are fairly technical, and its detailed description goes beyond the aims of this paper. As we stated before in the manuscript, the use marks were analysed using “an Olympus DSX10-UZH optical microscope, alternating the use of three lenses (DSX10-SXLOB plan 1x/0.03; DSX10-SXLOB plan 3x/0.09; DSX10-XLOB plan FL 10x/0.30) depending on the type of use marks”. We also specified that “the resulting images [that is, the 3D scans] have been produced with the Olympus DSX1000 software (v. 1.1.5.13)”. Like most modern microscopes, scales and margin errors are automatically considered and corrected by the software. The software's specific functions and options should deserve a separate publication, and they cannot be included in the present work. Finally, we replaced the unclear “equal distances” with the more precise “regular intervals”.

Lines 240-241. Describe these categories of intensity, and how they are recognised/defined.

We added here some references about the intensity of retouch, and we moved this sentence up in the section, together with the description of the other stigmata. Thank you. 

Line 257. I am unsure why bone retouching was necessary for this study, as the stigmata on bone retouchers have been extensively described and analysed in prior archaeological and experimental studies. It does not appear that anything new emerged from this exercise for the bone tools, particularly since the results were analysed using the conventions reported previously in these various studies. Table 4 seems superfluous to the aim of the study.

Thank you for your comment. It is good practice to have a “control group” in any experiment. In our case, we believe our study results more reliable if the stigmata on boomerangs and bone retouchers are analysed with the same methodology (e.g., 3D scans). Please refer to specific comments above for our response regarding the relevance of the parallel with bone retouchers. Finally, we address the issue on Table 4 in the comment below. Thank you. 

Line 268. How were blows counted? This is not described in the methodology. Was this determined by watching the videos, or by some other method?

That seems like an unusual question. As the leading author of the study, EFM was present during all the retouching sessions, and she had the role of supervising and documenting the experimental sessions. That includes counting the blows. Indeed, videos were watched afterwards to double-check the number of blows and the duration of the retouch activity, but we do not believe this level of detail is necessary for this context. Thank you.

Table 4. The various terms in the footnote of the table need to be described, as it is not immediately obvious what they refer to. However, many or most appear to be not particularly relevant to the aim of the manuscript, which is about stigmata on wood indentors from retouching. The paper is not about the stones per se, and indeed, they do not figure in any substantive way in the discussions or conclusions. The table could be made much simpler, leaving out most of this data.

Thank you for your comment. We gave this information in the Methods section and in the SI: “Each retouched flake was analysed after the retouching sessions. The modified edges were described following [39]: the considered features concern the position of the retouch, the delineation of the lithic edge after the retouch, the morphology of the detachments, the distribution of the retouch on each edge, the extent of the detachment scars on the flake’s surface, and the edge-angle of the lithic margin after the retouch (S1 Fig 3)”. As a general note, we believe every experimental study should have a robust multidisciplinary component. Our study investigates the technological implications of using boomerangs to retouch lithic tools: it would be incomplete if we did not show data on the lithic tools themselves. Moreover, it would make it difficult for lithic specialists to use our study to conduct their own research on retouch if data on the retouched lithic tools were missing.

Lines 289-290. Again we see something about the intentions of a knapper (see also comment for line 191). In this case, the intent was to remove long, thin flakes, which seems different from seeking to produce a certain type of edge. That is, the focus is on detaching certain types of flakes rather than creating an edge morphology. This seems more akin to core reduction. Some clarification/discussion is necessary, probably in the methodology section.

In order to streamline the text, we removed this section, and we integrated some of these information in SI. Thank you.

Line 292. What is meant by a ‘stepped in morphology’? Does this mean the flake scars ended in step terminations? This doesn’t fit comfortably with the description of the flakes as long and thin, combined with the retouched edges described as ‘semi-abrupt’ and ‘abrupt’. I also don’t know what is meant by ‘the retouch is located in a direct position’. More clarity is necessary.

In order to streamline the text, we removed this section and integrated some of this information into SI. Thank you.

Line 296. Here we read that the bone retouching gesture is ‘tangential’, the same as with the boomerang. Given the great difference in morphology of these indentors types, is it appropriate to use the identical term for the gesture(s) used in employing them? More clarity in description of the gestures, and variants of the gestures, is necessary.

We removed this specific section of the text and addressed the Reviewer’s question in a similar comment above. Thank you.

Line 297. What are ‘soft taps’, and how do these relate to the tangential ‘retouching movement’?

In order to streamline the text, we removed this section and integrated some of this information into SI. Thank you.

Line 321. ‘Struck’, not ‘stroke’

Thank you. We corrected accordingly.

Line 323. I don’t understand ‘the radius of the arch results in 7.4 mm on average’. Is this a thickness? By convention, it is good practice to list the standard deviation and sample size (i.e., 13 +/- __ mm, N=__).

We enriched this section in the Methods: “The width of the arc (C) and the height of the midpoint of the arc (H) were measured in correspondence to the use area. Those measurements are necessary to calculate the radius of the arch to assess the role that the convexity of the surface plays in the retouch activity. The radius of the arch was calculated as follows: H/2+C*2/8*H [36]”. Moreover, the sample size can be found in Table 3. Finally, we added the standard deviation as requested. Thank you.

Line 324-325. Here is a bit more information on the gestures used in the bone retouching and the boomerang retouching, but it is still unclear to me. How are the two related exactly? How are they different/similar?

We addressed this issue in a similar comment above. Thank you.

Line 336. What is meant by ‘scaled on the left and right edges’? Define the terminology used, or at least identify the source for a definition, with page number. What is the convention used to identify left and right edges (e.g., proximal end up, dorsal surface up, or proximal end down, dorsal surface up, or some other convention)? This comment applies across much of the description of results. Alternatively, much of this information is not strictly germane to the research question, so might be deleted.

In order to streamline the text, we removed this section. Thank you.

Line 341. What does ‘cross-section in correspondence of stigmata’ mean? I think this could be simplified. This is not the normal convention for showing a cross section. The profile line across the object should be shown with tick marks and keyed to the cross section drawing.

We refer here to the morphology of the cross-section of the boomerangs (i.e., plano-convex or bi-convex, as we explained in the Methods). The drawing represents an approximation of the shape of the cross-section in the portion of the boomerang where the stigmata are located. As requested, we simplified this caption and the captions of similar figures. Thank you.

Line 358. Here is another reference to the knapping gesture, this time referring to a parallel trajectory and contrasting it with a tangential trajectory. If the movement is parallel to the edge, how is a flake detached? Again, I think a figure is necessary to untease these variations, rather than describing them in words.

We addressed this issue in a similar comment above. Thank you.

Line 381. ‘the retouch is considered completed’. This suggests an intention on the part of the flintknapper, which should be described in the methodology.

We integrated the methodology accordingly. Retouching session 1: “The operators aimed to produce flakes that they felt might be functional for cutting or scraping activities, inspired by retouched tools from Palaeolithic contexts [e.g., Discoid and Quina Mousterian: 22,23, and references therein]”. Retouching session 2: “The operator aimed to produce flakes potentially functional for cutting or scraping activities, inspired by retouched tools from Australian archaeological contexts [e.g., 29]”. Thank you.

Line 389. Again, this refers to an unspecified goal on the part of the flintknapper. This needs to be discussed in the methodology.

Please refer to the comment above. Thank you.

Line 466 and thereafter. Rephrase to remove the word ‘goes’. Perhaps ‘varies’ is a better word to use here. Rote data descriptions like this are best put into summary tables rather than described in text.

We adjusted the wording in this section. Thank you for your suggestion.

Lines 475-476. The authors assert empirical differences between boomerangs and bone retouchers. This must be accompanied by statistical tests. If there appears intuitively to be a difference, then test this with statistics. All the data is there I think, so it shouldn’t be an onerous task.

The data is represented in S9 Fig 7, and this representation is purely visual (not statistical). We modified this figure by setting the length of the y-axis at the same size for boomerangs and retouchers, hoping to make the comparison clearer and more consistent. We would like to state that we did perform statistical tests of these data; however, because of the small size of the sample (17 use areas on boomerangs, 22 on bone retouchers), statistical results are not reliable. In order to clarify, we enriched the caption of S9 Fig 7 with the values resulting from the statistical study (i.e., correlation coefficient and p-value). Thank you.

Lines 480-482. The authors discuss partial differences in lengths and widths (not sure what that means), but greater differences in surfaces and perimeters, but surely the perimeters are a function of the lengths and widths? This whole discussion is important and should be expanded and documented using statistical tests. (This comment applies for most of the analysis which follows—statistical tests are appropriate to support the various assertions).

Please see the comment above regarding the requested statistical test. Again, this section is purely descriptive of the use areas produced on boomerangs. The only claim we make from this section is that we should not be limited to the maximum length and width of the use areas, but we should also investigate other metric data like perimeter and surface. To clarify, we did perform statistical tests on the relationship between surface and perimeter on the one hand, and length and width on the other. The results are not statistically relevant because of the small size of the sample, and for this reason, they are not presented here. The discrepancies between these measurements (which result in direct proportion in regular geometries) do not necessarily relate to one another in the context of irregular shapes like the analysed use areas. That represents how much several different variables influence the retouch activity. Thank you. 

Line 485. I don’t think the categories are defined anywhere. I do not understand how you can have concentrated and dispersed distributions superposed without it resulting in a concentrated distribution.

Thank you for your comment. The “intensity of retouch” categories refer to Mallye et al. (2012). We now specify it in the Methods. Concentrated and isolated distribution of the marks never appears together in the same use area. The meaning of this sentence is: “of all the analysed use areas on bone retouchers, 36% have a concentrated distribution of the marks, 32% a dispersed” etc. In order to clarify, we associated the real numbers with the percentages.

Line 493. The authors suggest that the retouching intensity on the retouchers varied according to the flintknapper’s techniques, but the cited data (S9 Fig 2c) don’t support this conclusion because the data is not comparable. It is possibly that YLP’s technique varied from TRM’s because YLP used both types of retouchers, and TRM used only bone retouchers. The variation is very likely related to the radically different nature of these types of retouchers, and the adjustments to gesture necessary to make them work properly. That is, they might have used identical gestures, but TRM didn’t have to adjust his because they didn’t take part in the boomerang experiment. I also note that YLP was naïve to using boomerang retouchers prior to the experiment, which is a complicating factor when attempting to characterise YLP’s technique.

As stated in the caption, this plot only takes into account data on the bone retouchers, not on the boomerangs. Moreover, retouching session 1 was carried out before retouching session 2, so the use of boomerangs could have not influenced the use of bone retouchers. Thank you.

Line 562. Is it possible that some of these marks (e.g., a-d) were created during the manufacture of the boomerangs using wood rasps and mechanical/abrasive power tools?

Thank you for your comment. As we stated in a similar comment above, it is general practice before starting an experimental protocol to analyse all the surfaces involved in the experiment, including manufacturing traces. After the experiment, we determined that none of the retouch-induced marks could be confused with the manufacturing traces observed before the retouching session, and for this reason, it is superfluous to address this topic in the manuscript.

Line 576. I note that Mallye et al. (2012: Fig 2) identified ‘scaled areas’ among pits and scores in their bone retoucher experiments that seem analogous to the ‘small detachments’ identified by the authors on the boomerang retouchers. This might be a more likely correlation than the peeling marks cited, and it should be mentioned and discussed. See also lines 737-738.

Thank you for your suggestion. The scaled areas defined by Mallye et al. (2012) are a relatively big portion of the surface, composed of several narrow, roundish depressions, generally associated with punctiform impressions. The marks identified in our study are single ‘detachments’ (not a group of stigmata), deeper than the ‘pitted areas’, and associated with linear impressions. For the sake of correctness, we cited Mallye et al. (2012) in the parts indicated by the reviewer.

Lines 579-583. This is not necessary here, as a whole section on this appears from Line 600. Line 585. Again, lots of assertions based on data, but no statistical tests to back up these intuitive assessments. Some statistical tests are warranted here.

We deleted the redundant sentence regarding the micro-flakes. Moreover, we slightly rephrase this section to make our calculations clearer. Finally, we added three scatter plots in S9 Figs 8-10 representative of statistical tests. Thank you.

Line 586. The authors should move more of this important data out of the supplementary information and into the main text. This whole section, for instance, is carried by data in the supplementary information. Supplementary information should be for supplementary data or raw data, not for the data which is key to the argument.

We are grateful to the reviewer for their suggestion. We decided to keep the statistical outcomes in the SI because doing otherwise would have meant submitting a manuscript with more than 30 figures. For brevity, we decided to prioritise the figures representing the outcomes of the traceological analysis because we believe it is the main core of our study.

Line 625-627. I strongly dispute this assertion. Ethnographic and anthropological accounts of boomerangs in Australia frequently describe how boomerangs are used for many, many things, both functionally and ritually. I would go so far as to suggest that ethnographic accounts of boomerangs in traditional societies almost always described (or implied) how they functioned in various roles, not just for throwing at animals. Archaeologists have also been explicit about the use of boomerangs for stone tool manufacture (e.g., Moore 2004, and possibly some of the earlier 20th century research on tool types). It would appear that this paragraph is a way of positioning or promoting the authors’ research, which is not inherently a bad thing, but it should be done thoughtfully and self-reflexively, and not by ignoring decades of prior research.

Thank you for your comment. We toned down and expanded this section to clarify our point: “The multipurpose nature of boomerangs is a well-established concept in Australian Aboriginal Traditional knowledge, although it has received little direct investigation by archaeologists and ethnographic researchers [e.g., 2,6–9,21]. In our recent work [2], we performed a systematic quantitative review of the literature available on the subject of “boomerangs”. Our analysis showed how most previous boomerang-focused publications mainly consider the aerodynamic properties linked to the boomerang’s infamous returning abilities. Technological and functional aspects of non-returning boomerangs -- including their use as retouchers -- can be found in the form of incidental citations within broader descriptions of the Indigenous Australian lifestyle and daily activities (e.g., woodworking) [2]. Consequently, the remarkable variability of non-returning boomerangs is frequently overlooked.

During the second half of the 20th century, ethnographic reports on boomerangs are present in the literature, but they seem to stagnate in approaches and references belonging to the beginning of the century. Although their contribution to the current knowledge on boomerangs has to be recognised, it could also be argued that they are nowadays less suitable to contribute to a multidisciplinary and more scientific approach to ancient technology [2]. Suffice it to know that Davidson’s work from 1936 [45] currently remains the main comprehensive study on boomerangs. Davidson himself concluded his contribution (p. 90) by wishing for an approach to boomerangs that was more technology-inclined than typological [2,45].

Regarding recent contributions boomerangs have been rarely studied: this is a relevant issue, considering how much theories and techniques for the study of ancient technologies evolved in the last 30 years. Nevertheless, in this timeframe boomerangs have only been the object of sporadic classifications [7,25,46]; in other cases, they played a marginal role in valid experimental programs [21,46]; finally, they were summoned when rare archaeological remains of boomerangs were discovered [15,16]. This approach has overshadowed the role that these tools have played, and keep playing, in Australian Indigenous societies”.

This section is complementary to the integrations we made to the Introduction (please see specific comment above). Thank you.

Lines 636-639. This succinct summary of the method is just appearing here for the first time. I am a little confused at how the descriptions of the retouched edges related to ‘functionality’, because the latter term is not defined. Boomerangs can be used to retouch, but we know this from ethnographic reports spanning over 100 years. So what is this study attempting to achieve exactly? I also note that the flintknapper’s feedback is not presented systematically, and there is nothing in the methodological description about how this feedback was collected. And traceological similarities between bone and wood doesn’t inform us about functionality per se, depending on how that is defined.

We moved the definition of functionality in the methods. The issue regarding the existing relationship between our experiment and literature evidence has been addressed in a specific comment above. Thank you.

Line 641. Here we have something more explicit on the intent of the flintknappers. This should appear in the methodology, not the conclusions. How did the flintknappers differentiate between the retouched Australian tools, and the Discoid/Quina Mousterian tools? Was there a perception of different functional criteria for the cutting/scraping tools in these two industries? This is important for discerning when the flintknapper decided a tool was ‘finished’, which in turn relates directly to the nature and extent of the stigmata distribution on the boomerang retouchers in particular.

We moved this information in the Methods as suggested. We addressed the issue regarding the intentions of the knappers in a specific comment above. Thank you.

Line 655-657. Something new appears here re the experimental protocol, about the spatial distribution of wear and the knapper’s gesture(s). This should have been in the methodology.

We moved this information in the Methods. Thank you.

Line 675. The traceological section is the strongest part of the paper, and has the most to contribute. I suggest that with this section in mind, the authors should revise the introduction and methodology to lead more seamlessly to this discussion. Some of the weaker parts of the manuscript can be deleted, with a clear and single-minded focus on the conclusions that appear in this section.

Thank you for your suggestion. We rephrased Introduction and Methods accordingly.

Line 681. The authors say that their data matches prior research on bone retouchers, but this is not surprising as they already concluded this at the outset of the study (Lines 224- 225).

We addressed this issue in a specific comment above. Thank you.

Line 707. Typo

We corrected the typo. Thank you.

Lines 707-709. This is good, but how applicable is it to Australia, where archaeological boomerangs are exceptionally rare and (probably) too poorly preserved to retain evidence of retouching? Identifying boomerang retouchers is most relevant to ethnographic collections in museums, unless their use can be related more specifically to the stone artefacts, as Moore (2004) has done in identifying soft-hammer retouched tulas.

We addressed the issue of the relevance of our study in the context of a debate on the role of wooden tools in the past; please refer to specific comment above. Moreover, we added a small section in the “Future research” section regarding the possibility of integrating the study of wooden retouchers with the features of retouched lithic edges. Thank you.

Line 764. The authors claimed to have ‘minimised the variables’ but this isn’t discussed explicitly in the methodology. What variables were minimised, and how was this accomplished?

In order to clarify, we added the following: “The main objective was to assess the functionality of boomerangs as retouchers. We attempted to minimise the variables -- e.g., using only one type of lithic raw material, entrusting the retouching only to expert knappers, using similar types of hardwood for manufacturing boomerangs, and using only two shapes of boomerangs. This approach allowed a clearer identification of which parameters are involved in the production of stigmata on the wooden surface and the relationship with the percussion movements characteristic of the retouching technique. This choice was also dictated by the scarcity of information on wooden retouchers and to what degree their response to percussion stimuli is comparable to that observed on osseous materials”. Thank you.

 

6. PLOS authors have the option to publish the peer review history of their article (what does this mean?). If published, this will include your full peer review and any attached files.

Do you want your identity to be public for this peer review? For information about this choice, including consent withdrawal, please see our Privacy Policy.

Reviewer #1: Yes: Luc Doyon

Reviewer #2: No

---

## [Decision Letter · Decision Letter 1]

3 Aug 2022

Beyond the main function: An experimental study of the use of hardwood boomerangs in retouching activities

PONE-D-22-05291R1

Dear Dr. Martellotta,

We’re pleased to inform you that your manuscript has been judged scientifically suitable for publication and will be formally accepted for publication once it meets all outstanding technical requirements.

Kind regards,

Enza Elena Spinapolice, Ph.D

Academic Editor

PLOS ONE

Additional Editor Comments (optional):

Reviewers' comments:

Reviewer's Responses to Questions

**Comments to the Author**

1. If the authors have adequately addressed your comments raised in a previous round of review and you feel that this manuscript is now acceptable for publication, you may indicate that here to bypass the “Comments to the Author” section, enter your conflict of interest statement in the “Confidential to Editor” section, and submit your "Accept" recommendation.

Reviewer #1: All comments have been addressed

Reviewer #2: All comments have been addressed

2. Is the manuscript technically sound, and do the data support the conclusions?

Reviewer #1: Yes

Reviewer #2: (No Response)

3. Has the statistical analysis been performed appropriately and rigorously? 

Reviewer #1: Yes

Reviewer #2: (No Response)

4. Have the authors made all data underlying the findings in their manuscript fully available?

Reviewer #1: Yes

Reviewer #2: (No Response)

5. Is the manuscript presented in an intelligible fashion and written in standard English?

Reviewer #1: Yes

Reviewer #2: (No Response)

6. Review Comments to the Author

Reviewer #1: I am pleased to see the modifications made to the original version of the manuscript entitled "Beyond the main function: An experimental study of the use of hardwood boomerangs in retouching activities" by Eva Francesca Martellotta and colleagues. The authors have provided satisfactory responses to the issues raised by the two reviewers, and either made the appropriate changes to the manuscript or provided a detailed, coherent and justified rebuttal in the rare instances where the requested changes were outside the scope of their study.

I fully support the publication of this manuscript in PLOS ONE as it provides new data on Australian Indigenous technological repertoire and has far-reaching implications on the study of Pleistocene stone knapping material culture.

The manuscript is well-structured and more streamlined compared to the original version. I once again congratulate the authors for their impressive investigation and for their dedication in sharing ALL their data which will allow sound comparisons in the future.

Before sending the manuscript to the processing team, the authors may want to consider the following aspects:

1) The term use wear is written in two ways, either with or without a dash. I would recommend to make it uniform throughout the text.

2) The word stigmata is used to refer to traces of use wear. This noun, both American and British English, refers either to a feeling, a flower part, or bodily marks related to wounds. I am aware its introduction in archaeological and use wear literature is in part the result of the direct translation of the French word "stigmates" but I believe it would be preferable to use more adequate terms, such as impact traces, use wear traces, etc.

3) Lines 160-161: Femurs are included twice, once with the tibia and once as a stand-alone category.

4) Lines 172-173: remove the footnote for Table 1.

5) In the footnote of Table 3 (lines 331-332), replace "low dash and a crescent number" to "underscore symbol and ascending number".

6) The metric comparison of the traces of impact on bone and boomerang is very interesting. However, the authors interpret the differences in data as a result of size differences between the two types of knapping tools. I would just like to point out that one cannot exclude the possibility that, assuming the same force was used when knapping with both raw material, variations in mineral content and structure density between bone and hardwood could explain the difference reported here. Obviously, this is an aspect that would require further biomechanical analyses and it falls completely outside the scope of the present study. Whether the authors want to consider this remark is entirely up to them and, should they decide not to do so, it would not decrease in any way the merit of their study.

7) The formatting of the references needs a final review. The titles should not be capitalized unless for proper nouns, latin names should be in italic, page range for book chapters should be included, the doi appears twice for ref 43, etc.

Reviewer #2: (No Response)

7. PLOS authors have the option to publish the peer review history of their article (what does this mean?). If published, this will include your full peer review and any attached files.

Reviewer #1: **Yes: **Luc Doyon

Reviewer #2: No

---

## [Editor Report · Acceptance letter]

5 Aug 2022

PONE-D-22-05291R1 

Beyond the main function: An experimental study of the use of hardwood boomerangs in retouching activities 

Dear Dr. Martellotta:

I'm pleased to inform you that your manuscript has been deemed suitable for publication in PLOS ONE. Congratulations! Your manuscript is now with our production department. 

Kind regards, 

on behalf of

Dr. Enza Elena Spinapolice 

Academic Editor

PLOS ONE